# High-molecular-weight polymers from dietary fiber drive aggregation of particulates in the murine small intestine

Asher Preska Steinberg[1], Sujit S Datta[2], Thomas Naragon[1], Justin C Rolando[1], Said R Bogatyrev[3], Rustem F Ismagilov[1,3]*

[1]Division of Chemistry and Chemical Engineering, California Institute of Technology, Pasadena, United States; [2]Department of Chemical and Biological Engineering, Princeton University, Princeton, United States; [3]Division of Biology and Biological Engineering, California Institute of Technology, Pasadena, United States

**Abstract** The lumen of the small intestine (SI) is filled with particulates: microbes, therapeutic particles, and food granules. The structure of this particulate suspension could impact uptake of drugs and nutrients and the function of microorganisms; however, little is understood about how this suspension is re-structured as it transits the gut. Here, we demonstrate that particles spontaneously aggregate in SI luminal fluid ex vivo. We find that mucins and immunoglobulins are not required for aggregation. Instead, aggregation can be controlled using polymers from dietary fiber in a manner that is qualitatively consistent with polymer-induced depletion interactions, which do not require specific chemical interactions. Furthermore, we find that aggregation is tunable; by feeding mice dietary fibers of different molecular weights, we can control aggregation in SI luminal fluid. This work suggests that the molecular weight and concentration of dietary polymers play an underappreciated role in shaping the physicochemical environment of the gut.

**Editorial note:** This article has been through an editorial process in which the authors decide how to respond to the issues raised during peer review. The Reviewing Editor's assessment is that all the issues have been addressed (see decision letter).

DOI: https://doi.org/10.7554/eLife.40387.001

*For correspondence:
rustem.admin@caltech.edu

## Introduction

The small intestine (SI) contains numerous types of solid particles. Some of these particles include microbes, viruses, cell debris, particles for drug delivery, and food granules (*Donaldson et al., 2016*; *McGuckin et al., 2011*; *Maisel et al., 2015a*; *Goldberg and Gomez-Orellana, 2003*; *Faisant et al., 1995*). Little is understood about the state of these particles in the small intestine; do these particles exist as a disperse solution or as aggregates? An understanding of how particulate matter is structured as it moves through the SI would contribute to fundamental knowledge on a host of topics, such as how microbes, including probiotics and pathogens, function in the SI (*Millet et al., 2014*; *Lukic et al., 2014*; *Del Re et al., 1998*; *Kos et al., 2003*; *Tzipori et al., 1992*). Knowledge of how particle suspensions change during transit would also provide insight into how the uptake of drugs and nutrients are affected by the physicochemical properties of the SI environment (*Maisel et al., 2015a*; *Goldberg and Gomez-Orellana, 2003*). It would also give us better comprehension of how the SI acts to clear potential invaders and harmful debris (*McGuckin et al., 2011*; *Howe et al., 2014*).

Polymers abound in the gut in the form of secretions (e.g. mucins and immunoglobulins) and dietary polymers (e.g. dietary fibers and synthetic polymers). It is well known that host-secreted polymers can cause aggregation of particles via chemical interactions; for example, mucins (***Puri et al.,***

*2015*; *Laux et al., 1986*; *Sajjan and Forstner, 1990*; *Wanke et al., 1990*; *Sun et al., 2007*), immunoglobulins (*Doe, 1989*; *Peterson et al., 2007*; *Levinson et al., 2015*; *Hendrickx et al., 2015*; *Endt et al., 2010*; *Bunker et al., 2017*; *Moor et al., 2017*; *Mantis et al., 2011*; *Donaldson et al., 2018*), and proteins (*Bergström et al., 2016*) can cause bacteria to aggregate via an agglutination mechanism. However, non-adsorbing polymers can also cause aggregation via purely physical interactions that are dependent on the physical properties of the polymers, such as their molecular weight (MW) and concentration (*Asakura and Oosawa, 1954*; *Asakura and Oosawa, 1958*; *Vrij, 1976*; *Gast et al., 1983*; *Prasad, 2002*; *Lu et al., 2006*; *Ilett et al., 1995*). Here, we investigate whether these physical interactions play a role in structuring particles in the SI. For this work, we study the interactions between polystyrene particles densely coated with polyethylene glycol (PEG) and the luminal contents of the SI. It has been demonstrated previously that PEG-coated particles have little or no chemical interactions with biopolymers (*Valentine et al., 2004*; *Wang et al., 2008*), so using PEG-coated particles allows us to isolate and investigate only the interactions dominated by physical effects.

## Results

### PEG-coated particles aggregate in fluid from the murine small intestine

It has been observed that both bacteria (*Levinson et al., 2015*; *Hendrickx et al., 2015*; *Endt et al., 2010*; *Moor et al., 2017*; *Donaldson et al., 2018*; *Bergström et al., 2016*) and particles (*Maisel et al., 2015a*; *Ensign et al., 2012*; *Tirosh and Rubinstein, 1998*; *Maisel et al., 2015b*) aggregate in the gut. Experiments have been performed in which mice are orally co-administered carboxylate-coated nanoparticles, which are mucoadhesive, and PEG-coated nanoparticles, which are mucus-penetrating (*Maisel et al., 2015a*). The carboxylate-coated particles formed large aggregates in the center of the gut lumen. In contrast, PEG-coated particles were sometimes found colocalized with carboxylate-coated particles and also penetrated mucus, distributing across the underlying epithelium of the SI as aggregates and single particles.

To evaluate the distribution of particulate suspensions in the SI, we suspended 1-μm-diameter fluorescent PEG-coated particles (see Materials and methods for synthesis) in buffers isotonic to the SI and orally administered them to mice. We chose 1-μm-diameter particles because of their similarity in size to bacteria. We collected luminal contents after 3 hr and confirmed using confocal fluorescence and reflectance microscopy that these particles aggregated with each other and co-aggregated with what appeared to be digesta (*Figure 1C and D*; Materials and methods). On separate mice, fluorescent scanning was used to verify that particles do transit the SI after 3 hr (*Figure 1A and B*; Materials and methods).

Given the rich complexity of the SI, wherein particles co-aggregate with digesta and bacteria, and are subjected to the mechanical forces of digestion and transit (*Hasler et al., 2009*), and other phenomena, we next developed an ex vivo assay to characterize the structure of particles in luminal fluid from the SI of mice. As a simple starting point, we sought to understand interactions among particles of known chemistry and the luminal fluid of the SI. To minimize chemical interactions with the biopolymers of the SI, we again chose PEG-coated polystyrene particles. PEG coatings have been shown to minimize biochemical interactions between polystyrene particles and biopolymers in a variety of contexts (*Valentine et al., 2004*; *Wang et al., 2008*), and thus PEG-coated particles are commonly used in drug delivery (*Maisel et al., 2015a*; *Maisel et al., 2015b*; *Lai et al., 2009*).

To create PEG-coated polystyrene particles for the ex vivo experiments, we took 1-μm-diameter carboxylate-coated polystyrene particles and conjugated PEG to the surface (Materials and methods). We used nuclear magnetic resonance (NMR) to verify that PEG coated the surface of the particles (see Materials and methods and *Table 1*). We found that by coating with PEG 5 kDa and then coating again with PEG 1 kDa to backfill the remaining surface sites on the particle allowed us to achieve a lower zeta potential than applying a single coat of PEG 5 kDa (*Table 1*). We chose these particles for use in our assay. It has been suggested in the literature that a near-zero zeta potential minimizes the interactions particles have in biological environments (*Wang et al., 2008*).

To collect luminal fluid from the SI of mice, we excised the SI of adult mice (8–16 weeks old), divided it into an upper and lower section, and gently collected the luminal contents on ice. To separate the liquid and solid phase, we centrifuged the contents and collected the supernatant. To

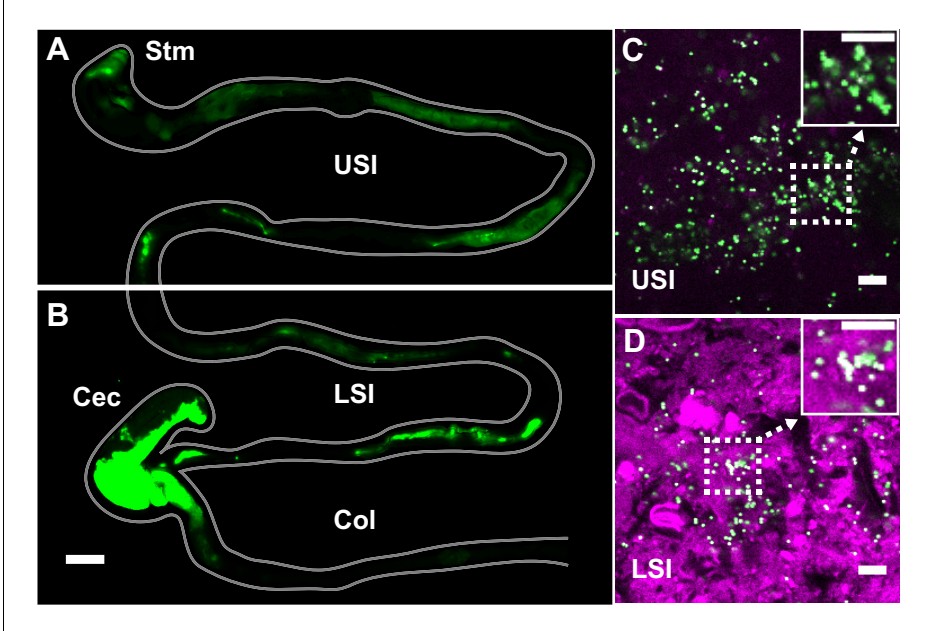

**Figure 1.** PEG-coated particles aggregate in the murine small intestine (SI). (**A and B**) Fluorescent scanner image of gastrointestinal tract (GIT) from a mouse orally administered a suspension of 1 µm diameter PEG-coated particles (green). Scale bar is 0.5 cm. (see *Figure 1—figure supplement 1* for image processing steps and how contours of gut were outlined). (**C and D**) Confocal micrographs of luminal contents from the upper (**C**) and lower (**D**) SI of a mouse orally gavaged with PEG-coated particles (green) showing scattering from luminal contents (purple). Scale bars are 10 µm. Stm = Stomach; USI = upper SI; LSI = lower SI; Col = colon.
DOI: https://doi.org/10.7554/eLife.40387.002
The following figure supplement is available for figure 1:

**Figure supplement 1.** Overview of image processing for fluorescent scanner images appearing in *Figure 1*.
DOI: https://doi.org/10.7554/eLife.40387.003

further ensure that any remaining solid material was removed from the fluid phase, we filtered the supernatant through a 30-µm pore size spin column and collected the filtrate (see Materials and methods for more details). We then placed the PEG-coated particles in the SI luminal fluid at a

**Table 1** Zeta potential and NMR measurements of PEG-coated particles.
For the zeta potential measurements, each particle solution was 0.1 mg/ml of particles in 1 mM KCl. Measurements were done on a Brookhaven NanoBrook ZetaPALS Potential Analyzer. Three trials were done where each trial was 10 runs and each run was 10 cycles. Values reported are the average zeta potential for the 30 runs. NMR measurements were performed as described in Materials and methods. Values are estimates of the nanomoles of polyethylene glycol (PEG) per milligrams of particles. To calculate this, we have to assume all the PEG on the surface is a single MW. It is therefore assumed all the PEG on the surface is PEG 5 kDa.

| Surface modification of PS particles | Zeta potential (mV) | Nanomoles PEG/mg particles |
|---|---|---|
| PEG 5 kDa | −18.87 ± 1.78 | 5.5 |
| PEG 5 kDa w/ mPEG 1 kDa backfill | −7.66 ± 2.12 | 4.6 |
| PEG 5 kDa w/ mPEG 350 Da backfill | −9.99 ± 1.65 | 4.3 |
| PEG 5 kDa w/ mPEG 5 kDa backfill | −14.56 ± 1.78 | 4.0 |
| PEG 2 kDa | −39.59 ± 2.41 | 9.4 |
| Carboxylate-coated (no PEG) | −61.36 ± 12.40 | 0.0 |

DOI: https://doi.org/10.7554/eLife.40387.004

volume fraction of ≈0.001. A low-volume fraction was chosen because bacteria in the healthy SI are found at similarly low-volume fractions (*Rubio-Tapia et al., 2009*; *O'Hara and Shanahan, 2006*; *Simon and Gorbach, 1984*). We found that, despite the PEG coating and low-volume fraction, aggregates of particles formed in 5–10 min (*Figure 2A–D*), a timescale much shorter than the transit time for food through the SI, which can be as short as ~80 min in healthy humans (*Hasler et al., 2009*) and ~60 min in mice (*Padmanabhan et al., 2013*). On longer timescales, peristaltic mixing could also play a role (*Hasler et al., 2009*); during fasting, the migrating motor complex (MMC) cycle first consists of a period of quiescence for ~30–70 min, followed by a period of random contractions, then by 5 to 10 min in which contractions occur at 11–12 counts per minute (cpm) in the duodenum and 7–8 cpm in the ileum. After eating, MMC is substituted with intermittent contractions in the SI and waves can occur at a frequency of 19–24 cpm in the distal ileum 1–4 hr later. We therefore chose to focus on aggregation at short timescales (~10 min) because we sought to understand

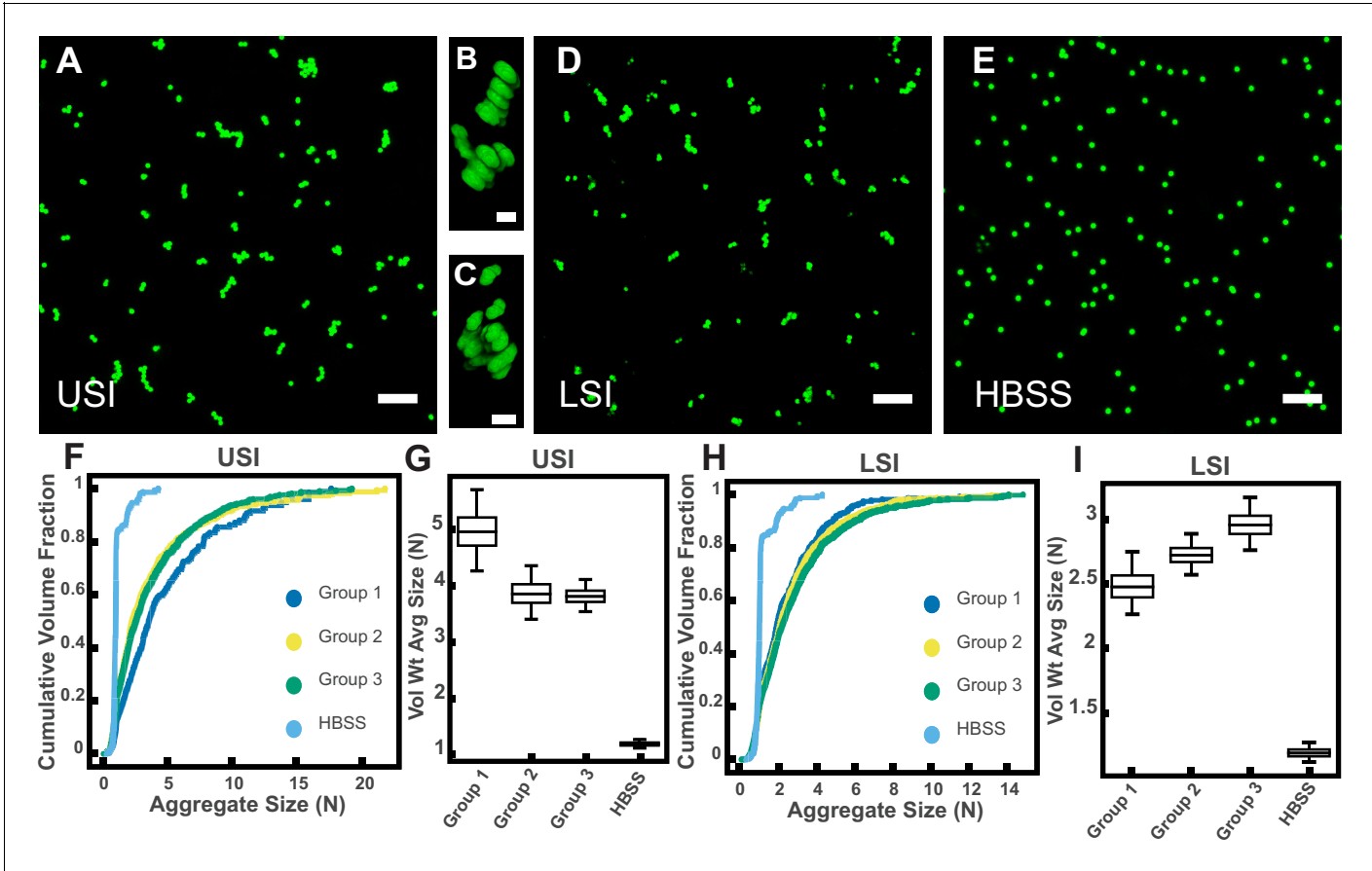

**Figure 2.** PEG-coated particles aggregate in fluid from the murine small intestine (SI) ex vivo. The 1-μm-diameter PEG-coated particles form aggregates in fluid collected from the upper (**A-C**) and lower (**D**) SI in ~10 min. (**A and D**) Maximum z-projections of 10 optical slices taken on a confocal microscope. (**B and C**) 3D renderings of aggregates found in panel A. (**E**) Maximum z-projection of the same particles in Hanks' balanced salt solution (HBSS). Scale bars are 10 μm in 2D images and 2 μm in 3D images. (**F and H**) Volume-weighted empirical cumulative distribution functions (ECDFs) comparing aggregation of the particles in pooled samples from the upper (**F**) and lower (**H**) SI of three separate groups of male chow-fed mice (each group consisted of three mice) and a control (particles suspended in HBSS). The vertical axis is the cumulative volume fraction of the total number of particles in solution in an aggregate of a given size. The horizontal axis (aggregate size) is given as the number of particles per aggregate (N). (**G and I**) Box plots depicting the 95% empirical bootstrap CI of the volume-weighted average aggregate size (given in number of particles per aggregate, N) in samples from the upper (**G**) and lower (**I**) SI (the samples are the same as those from panels F and H). The line bisecting the box is the 50th percentile, the upper and lower edges of the box are the 25th and 75th percentile respectively, and the whiskers are the 2.5th and 97.5th percentiles. USI = upper SI; LSI = lower SI. See Materials and methods for bootstrapping procedure.

DOI: https://doi.org/10.7554/eLife.40387.005

the initial formation of aggregates before aggregation is influenced by mechanical forces such as shear due to peristaltic mixing and the transit of food.

To quantify the amount of aggregation in samples of luminal fluid, we developed a method to measure the sizes of all aggregates in solution using confocal microscopy (see Materials and methods). From these datasets, we created volume-weighted empirical cumulative distribution functions (ECDFs) of all the aggregate sizes in a given solution. We used these volume-weighted ECDFs to compare the extent of aggregation in a given sample (*Figure 2F and H*). To test the variability of aggregation in samples collected from groups of mice treated under the same conditions, we compared the extent of aggregation in pooled samples taken from three groups, each consisting of three male mice on a standard chow diet. We plotted the volume-weighted ECDFs of each sample (*Figure 2F and H*) and observed that the variation among the groups under the same conditions appeared to be small compared with the differences between the samples and the control.

To quantify the variability of aggregation among groups using an additional method, we bootstrapped our datasets to create 95% bootstrap confidence intervals (CI) of the volume-weighted average aggregate size of each of the three groups and the control in Hanks' balanced salt solution (HBSS) (*Figure 2G and I*; see Materials and methods for complete details of the bootstrapping procedure). All 95% bootstrap CI either overlapped or came close to overlapping, again suggesting there was little variability among pooled samples treated under the same conditions (male mice on a standard chow diet).

## Fractionation of SI fluids suggests polymers play a role in aggregation of PEG-coated particles

Given that polymers can aggregate particles and bacteria via several mechanisms (*Puri et al., 2015*; *Laux et al., 1986*; *Sajjan and Forstner, 1990*; *Wanke et al., 1990*; *Sun et al., 2007*; *Doe, 1989*; *Peterson et al., 2007*; *Levinson et al., 2015*; *Hendrickx et al., 2015*; *Endt et al., 2010*; *Bunker et al., 2017*; *Moor et al., 2017*; *Mantis et al., 2011*; *Donaldson et al., 2018*; *Bergström et al., 2016*; *Asakura and Oosawa, 1954*; *Asakura and Oosawa, 1958*; *Vrij, 1976*; *Gast et al., 1983*; *Prasad, 2002*; *Lu et al., 2006*; *Ilett et al., 1995*), we hypothesized that biopolymers in SI luminal fluid are involved in the aggregation of the PEG-coated particles. We therefore sought to first quantify the physical properties of the polymers in the luminal fluid of the SI. To do this, we used a 0.45 μm filter to remove additional debris and ran samples from a group of three chow-fed mice on a gel permeation chromatography (GPC) instrument coupled to a refractometer, a dual-angle light scattering (LS) detector, and a viscometer (details in Materials and methods). Chromatography confirmed that polymers were indeed present in the SI fluid (*Figure 3A and D*). Because we do not know the refractive index increment (dn/dc) of the polymers present in these samples and the polymers are extremely polydisperse, we cannot make exact calculations of the physical parameters of these polymers. We can, however, calculate estimated values by assuming the range of the dn/dc values to be about 0.147 for polysaccharides and about 0.185 for proteins and then dividing the sample into different fractions based on retention volume (estimates of concentration and MW of polymers are displayed on *Figure 3A and D*). The estimates suggest that the SI is abundant in polymers with a range of MWs.

To qualitatively test our hypothesis that biopolymers in the SI were involved in the aggregation of our PEG-coated particles, we collected SI luminal fluid from a different group of three male, chow-fed mice. We performed an additional filtration step (0.45 μm) to further ensure the removal of any solid materials. This filtrate was then separated into aliquots and each aliquot was run through a different MW cut-off (MWCO) filter (see Materials and methods). We then collected the eluent of each aliquot and compared the aggregation of our PEG-coated particles in each (*Figure 3B,C,E and F*). We generally found less aggregation in the fractionated samples compared with the 30- and 0.45 μm filtered samples. When the MWCO was decreased to 3 kDa, the observed aggregation in the eluent matched the extent of aggregation observed for particles in HBSS. Overall, these data supported our hypothesis that polymers were involved in the aggregation of these particles.

Interestingly, in the lower SI, we observed more aggregation in the 0.45 μm filtered sample compared with the 30 μm filtered sample. From handling the samples, we observed that the 30 μm filtered samples appeared to be more viscous than the 0.45 μm filtered samples. We postulate that this increase in viscosity was due to the formation of self-associating polymeric structures, although we did not test this assumption. We attribute this decrease in aggregation in the 30 μm filtered

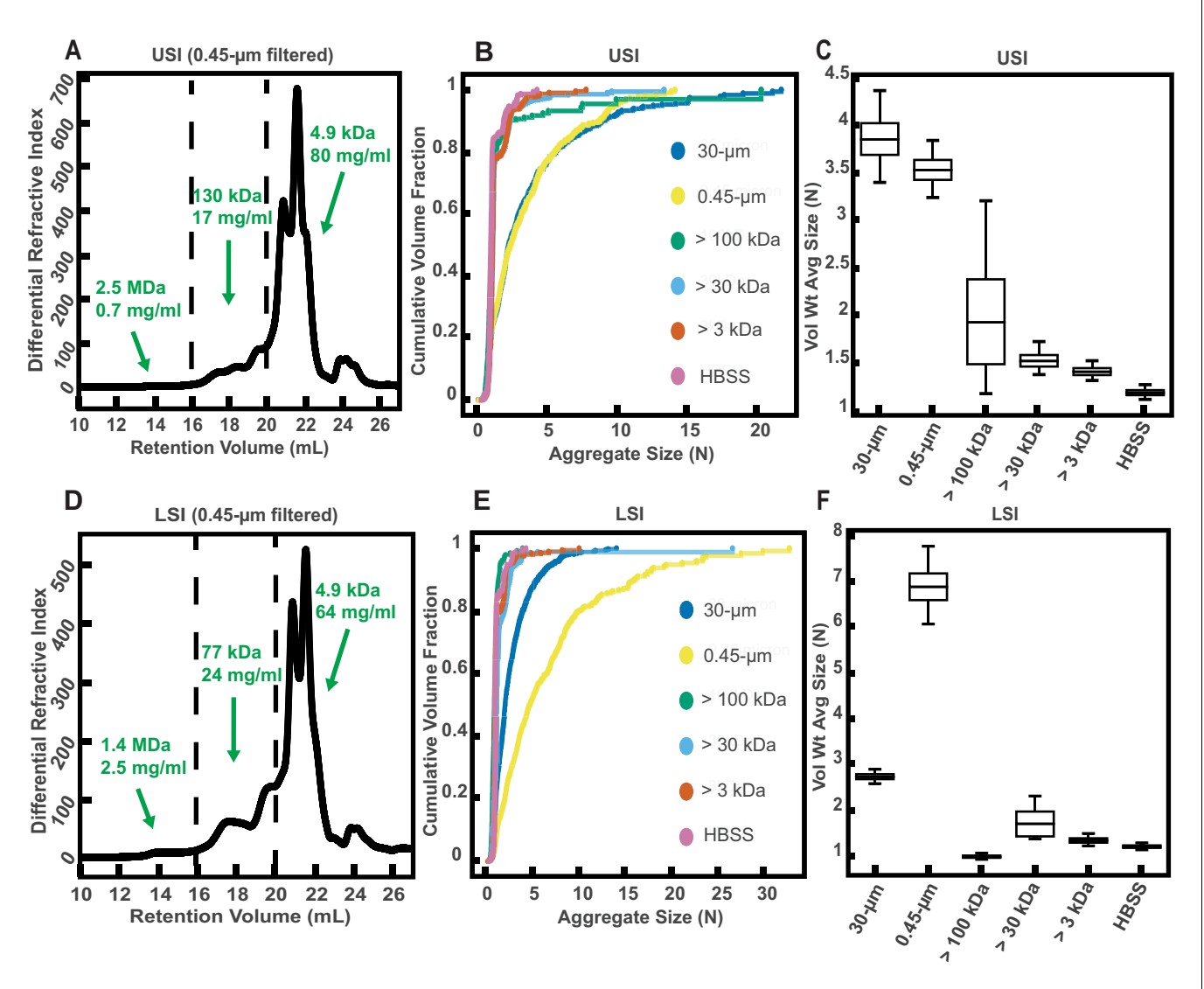

**Figure 3.** Gel permeation chromatography (GPC) of fluid from the small intestine (SI) and aggregation of PEG-coated particles in fractionated fluid from SI. (A and D) Chromatograms of samples from the upper (A) and lower (D) SI from a group of three chow-fed mice. Dashed lines indicate the three retention volumes the chromatograms were divided into for analysis: 11–16 mL, 16–20 mL, and >20 mL. Estimated concentrations and molecular weight (MW) are reported in green on the chromatograms for each retention volume. (B and E) Volume-weighted empirical cumulative distribution functions (ECDFs) of aggregate sizes in the upper (B) and lower (E) SI liquid fractions of chow-fed mice which have been run through MW cut-off (MWCO) filters with different MWCOs. As a control, aggregate sizes were also measured for particles placed in HBSS. The vertical axis is the cumulative volume fraction of the total number of particles in solution in an aggregate of a given size. The horizontal axis is aggregate size (number of particles per aggregate, N). (C and F) Box plots depict the 95% empirical bootstrap CI of the volume-weighted average aggregate size (given in number of particles per aggregate, N) in the samples from panels B and E, respectively (see Materials and methods for bootstrapping procedure). The line bisecting the box is the 50th percentile, the upper and lower edges of the box are the 25th and 75th percentile respectively, and the whiskers are the 2.5th and 97.5th percentiles.

DOI: https://doi.org/10.7554/eLife.40387.006

samples to slower aggregation kinetics due to decreased diffusivity of particles in this viscous medium. This decrease in aggregation at high polymer concentrations or viscosities is also observed in solutions of model polymers, as discussed in the next section.

# Aggregation of PEG-coated particles in model polymer solutions shows complex dependence on the concentration and MW of polymers

Before exploring the complex environment of the SI further, we sought to first understand how our PEG-coated particles behaved in simple, well-characterized polymer solutions with similar MW and concentrations to those polymers we found in the SI in the previous experiments (*Figure 3A and D*). It has been demonstrated that the aggregation of colloids and bacteria can be controlled by altering the concentration and size of the non-adsorbing polymers to which particles are exposed (*Asakura and Oosawa, 1954*; *Asakura and Oosawa, 1958*; *Vrij, 1976*; *Gast et al., 1983*; *Prasad, 2002*; *Lu et al., 2006*; *Ilett et al., 1995*). In these controlled settings, particles aggregate due to what are known as depletion interactions (*Asakura and Oosawa, 1954*; *Asakura and Oosawa, 1958*; *Vrij, 1976*). Many groups have focused on depletion interactions with hard-sphere-like colloids; they often use polymethylmethacrylate particles sterically stabilized with polyhydroxystearic acid, because these particles closely approximate hard-sphere-like behavior (*Royall et al., 2013*; *Pusey and van Megen, 1986*). In these scenarios, depletion interactions are often described as forces that arise when particles get close enough to exclude polymers from the space between them, resulting in a difference in osmotic pressure between the solution and the exclusion region, leading to a net attractive force (*Asakura and Oosawa, 1954*; *Asakura and Oosawa, 1958*; *Vrij, 1976*; *Gast et al., 1983*; *Prasad, 2002*). Others have instead chosen to describe the phase behavior of the colloid/polymer mixture in terms of the free energy of the entire system (*Ilett et al., 1995*; *Lekkerkerker et al., 1992*). Short-range attractions (polymer radius is ten-fold less than particle radius) between hard-sphere colloids induced by polymers have been described successfully in the language of equilibrium liquid–gas phase separation (*Lu et al., 2008*; *Zaccarelli et al., 2008*).

Some groups have explicitly accounted for the effects of the grafted polymer layer used to sterically stabilize colloids when studying interactions between polymer solutions and colloids (*Vincent et al., 1986a*; *Cowell et al., 1978*; *Vincent et al., 1980*; *Clarke and Vincent, 1981*; *Feigin and Napper, 1980*; *Vincent et al., 1986b*; *Gast and Leibler, 1986*; *Jones and Vincent, 1989*; *Napper, 1983*); this includes groups studying mixtures of polystyrene particles sterically stabilized with grafted layers of PEG (MWs of 750 Da and 2 kDa) and aqueous solutions of free PEG polymer (MW from 200 Da to 300 kDa) (*Cowell et al., 1978*; *Vincent et al., 1980*). It has been found experimentally that in mixtures of polymers and sterically stabilized colloids, the colloids form aggregates above a threshold polymer concentration. At even higher concentrations, as the characteristic polymer size shrinks, the colloids cease to aggregate, a phenomenon referred to as 'depletion stabilization.'

To test whether our PEG-coated particles behave similarly to what has been previously found in mixtures of polymers and sterically stabilized particles, we created polymer solutions with PEG at a range of polymer concentrations and MWs and measured the extent of aggregation in these polymer/particle mixtures (*Figure 4A–D*). We chose PEGs that have MWs similar to the MW of polymers we found naturally occurring in the SI (*Figure 3A,D*): 1 MDa, 100 kDa, and 3350 Da. Using PEGs with similar physical properties (i.e. MW, concentration) as a simple model of polymers found in the SI allows us to focus solely on physical interactions between the particles and polymers. We created PEG solutions in HBSS at mass concentrations similar to those measured for polymers in the SI (*Figure 3A and D*) and imaged the polymer/particle mixtures after ~10 min. HBSS was chosen because it has a similar pH and ionic strength to that of the SI (*Lindahl et al., 1997*; *Fuchs and Dressman, 2014*). At the high ionic strengths of these buffered aqueous solutions (~170 mM), any electrostatic repulsions that can occur between particles should be screened to length scales on the order of the Debye screening length ~0.7 nm (*Yethiraj and van Blaaderen, 2003*; *Jones, 2002*), nearly an order of magnitude smaller than the estimated length of the surface PEG brush (~6.4 nm; see Materials and methods for more details). We again chose to look at aggregation on short timescales (after ~10 min) because we sought to understand the initial formation of aggregates; in the SI, on longer timescales, aggregation will likely also be influenced by mechanical forces such as shear due to peristaltic mixing and the transit of food.

For PEG 1 MDa and 100 kDa solutions we found aggregates of similar sizes to those observed in the SI luminal fluid (*Figure 4A-D*). We did not detect any aggregation for the PEG 3350 Da solutions (*Figure 4D*). Because the pH is known to vary across different sections of the gastrointestinal tract and this could affect the observed aggregation behavior, we measured the pH in luminal fluid from

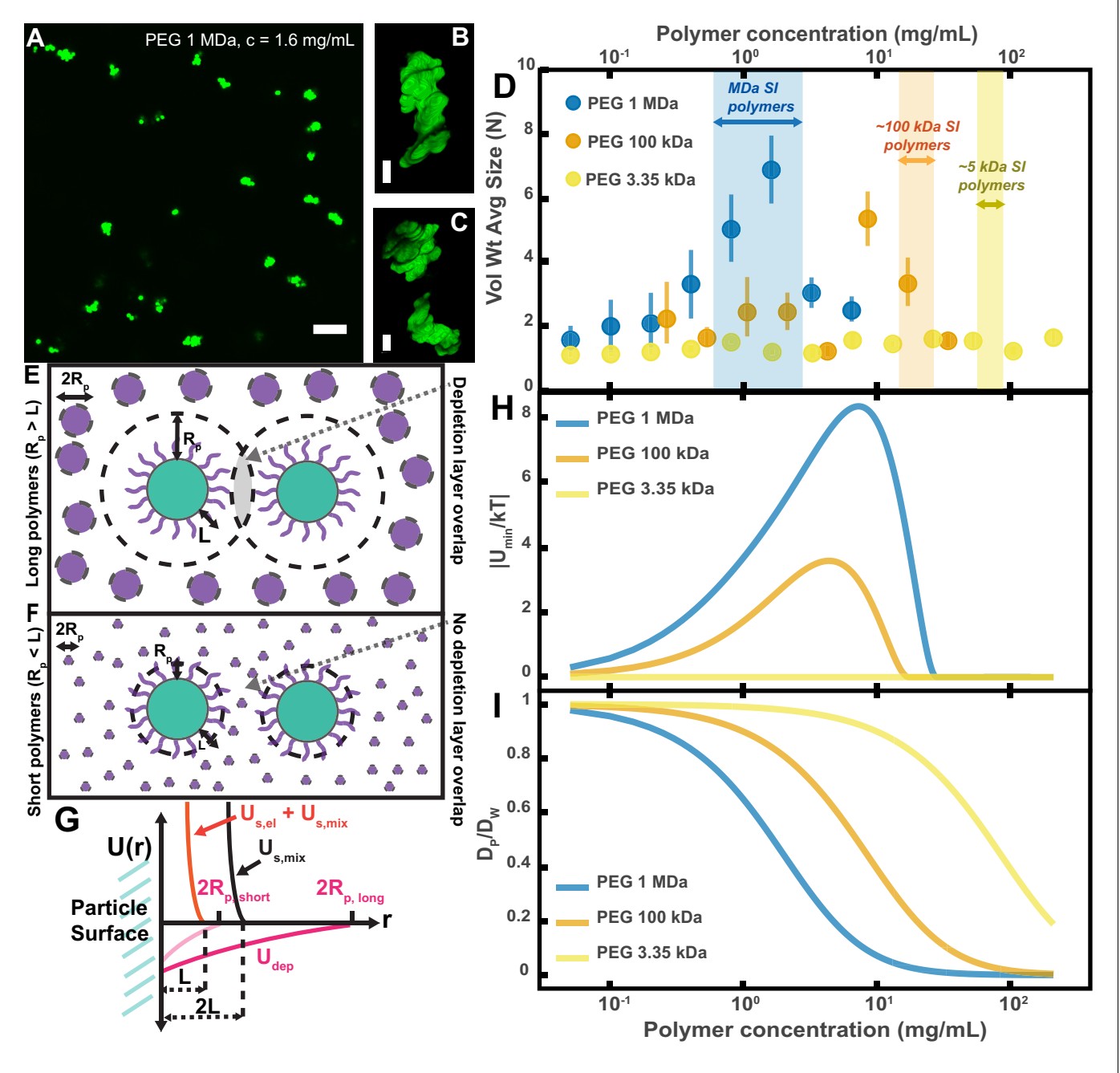

**Figure 4.** Aggregation of PEG-coated particles in model polymer solutions shows complex dependence on molecular weight (MW) and concentration of PEG. (A) Aggregates of 1 μm diameter PEG-coated particles in a 1 MDa PEG solution with a polymer concentration (c) of 1.6 mg/mL. Image is a maximum z-projection of 10 optical slices taken on a confocal microscope. Scale bar is 10 μm. (B and C) 3D renders of aggregates found in panel A. Scale bars are 2 μm. (D) Volume-weighted average sizes for serial dilutions of PEG solutions of three MWs (1 MDa, 100 kDa, and 3350 Da). Volume-weighted average sizes are plotted on the vertical axis in terms of number of particles per aggregate (N) against polymer mass concentration ($c_p$) in mg/mL. The vertical error bars are 95% empirical bootstrap CI (see Materials and methods for bootstrapping procedure). Shaded regions indicate the concentration ranges of detected intestinal polymers of similar MW. (E) Schematic depicting depletion interactions induced by 'long polymers' (polymer radius ($R_P$) >length of the brush, L). Free polymers are depicted as purple spheres. Colloids are depicted in green with the grafted brush layer in purple. The depletion layer around each colloid is depicted by dashed lines. The overlap region between the two depletion layers is indicated in grey. (F) Schematic depicting depletion interactions induced by 'short polymers' ($R_P$ <L). The depletion zone does not extend past the length of the brush and there is effectively no overlap in the depletion layers; the depletion attractions are 'buried' within the steric layer. (G) Schematic depicting the different contributions to the inter-particle potential (U(r)) against inter-particle separation distance (r). The hard surfaces of the particles are in contact at r = 0. $U_{dep}$ depicts the depletion potential for a short polymer ($R_{P,short}$) and a long polymer ($R_{P,long}$). $U_{s,mix}$ shows the contribution to the steric

*Figure 4 continued on next page*

*Figure 4 continued*

potential due to mixing. $U_{s,el} + U_{s,mix}$ shows the contribution due to elastic deformations and mixing at close inter-particle separations. (H) The magnitude of the minima of the inter-particle potential ($U_{min}/kT$) plotted against polymer concentration for the three PEG solutions in (D). (I) Diffusion coefficients estimated from the Stokes–Einstein–Sutherland equation for 1 µm particles in the PEG solutions used in (D). Diffusion coefficients of particles in polymer solutions ($D_P$) are normalized by the diffusion coefficients in water ($D_W$) and plotted against polymer concentration.

DOI: https://doi.org/10.7554/eLife.40387.007

The following figure supplements are available for figure 4:

**Figure supplement 1.** pH measurements of luminal fluid from different sections of the gastrointestinal tract.

DOI: https://doi.org/10.7554/eLife.40387.008

**Figure supplement 2.** Aggregation of PEG-coated particles in model polymer solutions with different pH.

DOI: https://doi.org/10.7554/eLife.40387.009

**Figure supplement 3.** Aggregation of PEG-coated particles in model polymer solutions from *Figure 4D* normalized by polymer overlap concentration.

DOI: https://doi.org/10.7554/eLife.40387.010

the upper and lower small intestine (see *Figure 4—figure supplement 1* and Materials and methods). We found that the upper small intestine (USI) luminal fluid was pH = 6.0 ± 0.1 and for the lower small intestine (LSI) pH = 7.5 ± 0.3. For the HBSS used, pH = 7.6 ± 0.1 (See Materials and methods), which matches that of the LSI but not the USI. We therefore conducted the same in vitro experiment for PEG 1 MDa in phosphate buffered saline with pH = 6.0 ± 0.1 (Materials and methods and *Figure 4—figure supplement 2*). We found some differences in the aggregation, but the overall trends were similar to before.

Overall, although our system is not at equilibrium at these short timescales, we found trends consistent with what has been observed in the literature for depletion interactions with sterically stabilized particles (*Vincent et al., 1986a*; *Cowell et al., 1978*; *Vincent et al., 1980*; *Clarke and Vincent, 1981*; *Feigin and Napper, 1980*; *Vincent et al., 1986b*; *Gast and Leibler, 1986*; *Jones and Vincent, 1989*; *Napper, 1983*). At dilute polymer concentrations, the extent of aggregation increased with concentration. At higher polymer concentrations, the extent of aggregation began to decrease as the solutions begin to 're-stabilize.' Additionally, the extent of aggregation was greater for longer polymers. Interestingly, we found that the curves for the long polymers in *Figure 4D* could be collapsed by normalizing the polymer concentration by the overlap concentration (which denotes the transition between the dilute to semi-dilute polymer concentration regimes) for each respective polymer solution (*Figure 4—figure supplement 3*). We next sought to describe the inter-particle potential using theory that combines depletion interactions with steric interactions.

We applied previously established theoretical frameworks that combine depletion interactions with steric interactions to better understand our system (*Vincent et al., 1986a*; *Feigin and Napper, 1980*; *Napper, 1983*). To account for the depletion attractions between colloids we used the Asakura–Oosawa (AO) potential ($U_{dep}$) (*Asakura and Oosawa, 1954*; *Asakura and Oosawa, 1958*; *Vrij, 1976*):

$$U_{dep}(r) = \begin{cases} +\infty \; for \; r \leq 0 \\ -2\pi\Pi_P a\left(R_P - \frac{r}{2}\right)^2 for \; 0 < r < 2R_P \\ 0 \; for \; r > 2R_P \end{cases} \tag{1}$$

Where $U_{dep}$ is given in joules, $\Pi_p$ is the polymer osmotic pressure (in Pa), $a$ is the radius of the colloid (in m), $R_p$ is the characteristic polymer size (in m), and $r$ is the separation distance between bare particle surfaces (in m). This form of the depletion potential equation assumes that $a \gg R_p$, a condition satisfied for 1 µm particles we used. For the polymer osmotic pressure, we used the following crossover equation for a polymer in a good solvent (*Rubinstein and Colby, 2003*; *Cai, 2012*):

$$\Pi_P = \frac{N_{Avo}kT}{MW} c_P \left(1 + \left(\frac{c_P}{c_P^*}\right)^{1.3}\right) \tag{2}$$

Where $\Pi_P$ is given in pascals, $N_{Avo}$ is Avogadro's number, $k$ is the Boltzmann constant, $T$ is the temperature (in kelvins), $MW$ is the molecular weight of the polymer (in Da), $c_p$ is the polymer mass concentration (in kg/m$^3$), and $c_P^*$ is the polymer overlap concentration (in kg/m$^3$). This equation describes the polymer osmotic pressure well in both the dilute and semi-dilute regime.

For the characteristic polymer size, we used the concentration-dependent radius of gyration (*Prasad, 2002*; *Burchard, 2001*). This can be written as:

$$R_P(c_P) = R_g(0)\left(\frac{Mw}{N_{Avo}kT}\frac{d\Pi_P}{dc_P}\right)^{-\frac{1}{2}}$$

(3)

Where $R_P(c_P)$ is the concentration-dependent radius of gyration or the characteristic polymer size given in meters, $R_g(0)$ is the radius of gyration (in m) at dilute concentrations and $\Pi_P$ is given by *equation 2*. The characteristic polymer size is given by the dilute radius of gyration at low concentration and is close to the correlation length of the polymer solution, or the average distance between monomers, in the semi-dilute regime. Therefore, using *equation 2 and 3*, we acquire the correct limits for the depletion potential; the Asakura–Oosawa potential in the dilute regime and the depletion potential described by Joanny, Liebler, and de Gennes in the semi-dilute regime (*Joanny et al., 1979*). Similar crossover equations have been found to adequately describe experimentally observed depletion aggregation in polymer-colloid mixtures where the polymer concentration spans the dilute and semi-dilute regimes (*Verma et al., 1998*). Using literature values for the hydrodynamic radii of the PEGs (*Armstrong et al., 2004*) and the Kirkwood-Riseman relation, which relates the hydrodynamic radius to the radius of gyration (*Armstrong et al., 2004*; *Tanford, 1961*; *Lee et al., 2008*), we estimated $R_g(0)$ for each polymer. We estimated $R_g(0) \approx$ 62.6, 16.7, 2.9 nm for PEG 1 MDa, 100 kDa, and 3350 Da, respectively. Using both the estimates of $R_g(0)$ and the MW of each polymer, we then estimated $c_p^*$ for each polymer (*Rubinstein and Colby, 2003*; *Flory, 1953*). We estimated $c_p^*$ = 1.6, 8.6, and 52.6 mg/mL for PEG 1 MDa, 100 kDa, and 3350 Da, respectively.

To account for steric interactions between the two grafted layers upon close inter-particle separations, we used *equation 4* (*Vincent et al., 1986a*; *Vincent et al., 1980*). For inter-particle separation distances between $L$ and $2L$, where $L$ is the length of the grafted layer, the steric interactions between the two grafted layers can be described using the Flory–Huggins free energy of mixing:

$$U_{s,mix}(r) = \frac{4\pi akT}{v_1}\left(\bar{\phi}_2^a\right)^2\left(\frac{1}{2}-\chi\right)\left(L-\frac{r}{2}\right)^2$$

(4)

Where $U_{s,mix}$ is the steric interaction energy due to mixing (given in joules), *a* is the particle radius (in m), $v_1$ is the volume of a water molecule (in m³), $\bar{\phi}_2^a$ is the average volume fraction of the grafted polymer (unitless), $\chi$ is the Flory–Huggins interaction parameter for the grafted polymer and the solvent (unitless), and $L$ is the length of the grafted layer (in m). For PEG in aqueous solvents, $\chi$ = 0.45 (*Brandrup et al., 1975*). Our NMR measurements (see Materials and methods for details) suggest that the grafting density of PEG is within the brush regime. We therefore use the Alexander–de Gennes approximation (*Rubinstein and Colby, 2003*) and our NMR measurements to estimate the length of the grafted layer ($L$) as $L \sim 6.4$ nm and the average volume fraction to be $\bar{\phi}_2^a \sim 0.43$.

For inter-particle separations closer than $L$, one needs to account for elastic deformations of the grafted layers (*Vincent et al., 1986a*; *Jones and Vincent, 1989*). This is far greater in magnitude than $U_{dep}$, so one can simply assume that at this point the potential is extremely repulsive. For inter-particle separations greater than $L$:

$$U(r) = \begin{cases} U_{s,\,mix} + U_{dep}\;for\;L < r < 2L \\ U_{dep}\;for\;r \geq 2L \end{cases}$$

(5)

Using this theoretical framework, we can build a physical intuition for the system (*Figure 4E–G*). Long polymers have depletion layers that extend out past the brush layer and overlap, inducing attractions between the particles (*Figure 4E*). For short polymers ($R_p < L$), the depletion attractions are buried within the steric repulsions induced by the brush and there are effectively no attractions among the particles (*Figure 4F*). We can use this crossover to estimate the magnitudes of the minima in the inter-particle potentials for the three PEG solutions (*Figure 4H*). It should be noted that we have made several simplifications; for example, we do not consider interactions between free polymers and the grafted layer, which could lead to partial penetration of free polymers into the grafted layer or possible compression of the grafted layer by the free polymers (*Vincent et al., 1986a*; *Gast and Leibler, 1986*; *Jones and Vincent, 1989*). Despite such simplifications, we find that the calculated minima display similar concentration trends to the trends seen in the average

aggregate sizes (*Figure 4D*). These calculations offer an explanation for why there is no aggregation of PEG-coated particles in solutions of PEG 3350.

Another factor that needs to be considered at the short timescales and low-volume fractions we are working at is aggregation kinetics (*Weitz et al., 1984*; *Ball et al., 1987*; *Smoluchowski, 1916*). The probability that particles collide in solution is directly related to the diffusion coefficient and the volume fraction of the particles. As we increase the polymer concentration we increase the viscosity of the solution and decrease the diffusivity of the particles. In *Figure 4I*, we plot theoretical estimates of the diffusion coefficients of the particles against the concentrations of the PEG solutions. These diffusion coefficients were estimated using literature measurements, the Stokes–Einstein–Sutherland equation, and the Huggins equation for viscosity (*Rubinstein and Colby, 2003*; *Armstrong et al., 2004*).

Because our system would not have reached equilibrium, in this case the non-monotonic dependence of aggregation on polymer concentration for long polymers is due to a complex interplay between thermodynamics and kinetics (which we have not untangled). However, both the dependence of diffusivity (*Figure 4I*) and the equilibrium prediction of inter-particle minima (*Figure 4H*) on polymer concentration suggest that we should expect a decrease in aggregation at high polymer concentrations. The inter-particle minima also suggest that we should not expect short polymers to induce aggregation. Both trends are consistent with what we observe. Understanding how our PEG-coated particles behave in these so-called 'simple' polymer solutions with similar physical properties to the intestinal polymers we detected (*Figure 3A and D*) informs the interpretation of the results of the next sections.

## MUC2 may play a role in the aggregation of PEG-coated particles, but is not required for aggregation to occur

It has been demonstrated that mucins can aggregate and bind to bacteria in vitro (*Puri et al., 2015*; *Laux et al., 1986*; *Sajjan and Forstner, 1990*; *Wanke et al., 1990*; *Sun et al., 2007*); thus, we wanted to test whether mucins, such as Mucin 2 (MUC2), which is the primary mucin secreted in the SI (*Ermund et al., 2013*; *Johansson et al., 2011*), drive the aggregation of PEG-coated particles in SI fluid. It is known that in the presence of $Ca^{2+}$ and at $pH \leq 6.2$, MUC2 can form aggregates or precipitate out, but it is soluble without $Ca^{2+}$ or at higher pH (*Ambort et al., 2012*). Our measurements of the pH throughout the SI suggest that it is possible that MUC2 precipitates out in the upper small intestine; however, because it is unclear how much $Ca^{2+}$ is in the lumen of the upper small intestine, there could be soluble MUC2 in the upper small intestine. Additionally, the literature suggests that, based on the pH, there should be soluble MUC2 in the lower small intestine. We therefore tested if MUC2 drives aggregation in both the upper and lower small intestine. To do this, we compared the aggregation of our PEG-coated particles in samples from MUC2 knockout (MUC2KO) mice to samples from wild-type (WT) mice. To carefully preserve the native composition of the SI fluid, we used a protease-inhibitor cocktail when collecting the samples (see Materials and methods). We confirmed mouse MUC2KO status via genotyping and Western blot (*Figure 5E*; Materials and methods). The Western blot detected MUC2 in the colons of WT mice and not MUC2KO mice, as expected, however it did not detect a signal for MUC2 in the SI of either the WT or MUC2KO mice. We speculate that the lack of MUC2 signal in the SI of WT mice may be due to low levels of MUC2 present in the luminal contents of the SI.

We observed aggregation in samples from both the MUC2KO and WT mice (*Figure 5A–B*). To test the strength of the aggregation effect in the different samples, we serially diluted the samples and measured the average aggregate size to see when the effect disappeared (*Figure 5C–D*). As explained in the previous section, we do not necessarily expect to see a linear decrease in aggregation with dilution. For simplicity, we will refer to the dilution factor at which aggregation begins to disappear as the 'aggregation threshold.' We found differences in the aggregation threshold in the samples from MUC2KO and WT mice (*Figure 5C–D*), suggesting that although MUC2 is not required for aggregation to occur, it could play a role in the aggregation of PEG-coated particles.

We wanted to test differences in the MW distribution of the polymers found in these samples, so we 0.45-µm-filtered our samples and analyzed them by GPC (see Materials and methods). The chromatograms from the refractometer (*Figure 5F–G*) suggest that the polymer composition of MUC2KO and WT samples were qualitatively similar. Following the same methods in *Figure 3*, we made estimates of the physical parameters of the detected polymers. These estimates are

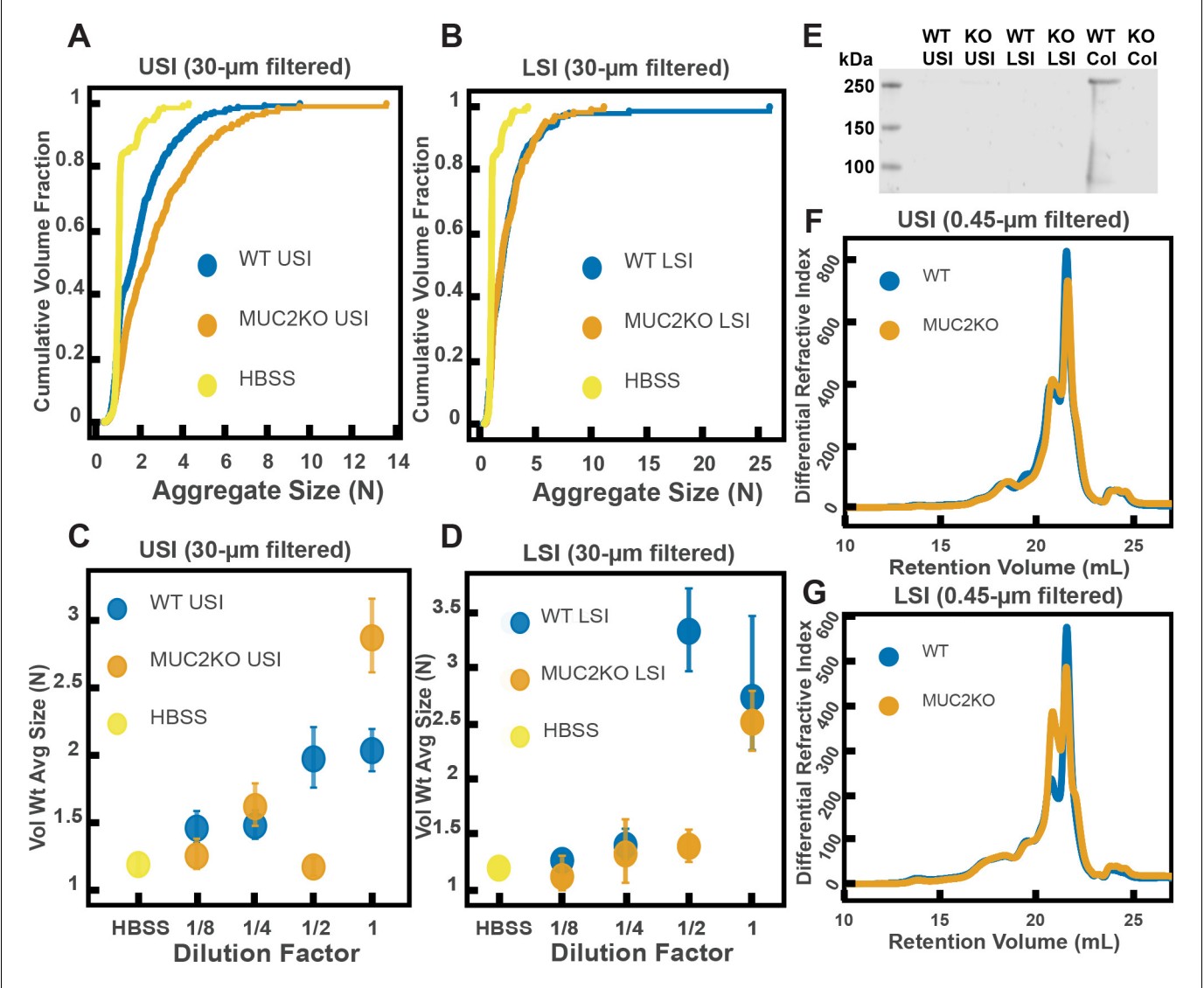

**Figure 5.** Quantification of the aggregation of particles in the small intestine (SI) in MUC2 knockout (MUC2KO) and wild-type (WT) mice. (**A and B**) Volume-weighted empirical cumulative distribution functions (ECDFs) comparing aggregation of the particles in undiluted, 30 μm filtered samples from the upper (**A**) and lower (**B**) SI of two separate groups of wild-type (WT) and MUC2-knockout (MUC2KO) mice to the control (particles suspended in HBSS). The vertical axis is the cumulative volume fraction of the total number of particles in solution in an aggregate of a given size; the horizontal axis is aggregate size in number of particles per aggregate (N). (**C and D**). Volume-weighted average aggregate sizes (Vol Wt Avg Size) for serial dilutions of 30-μm-filtered samples from the upper (**C**) and lower (**D**) SI of two separate groups of WT and MUC2KO mice. The dilution factor is plotted on the horizontal axis; a dilution factor of 1 is undiluted, ½ is a two-fold dilution. The vertical error bars are 95% empirical bootstrap CI (see Materials and methods). (**E**) Western blots of 30 μm filtered samples from the SI and the colon of WT and MUC2KO mice. WT USI = WT upper SI; KO USI = KO lower SI; WT LSI = WT lower SI; KO USI = KO upper SI; WT Col = WT colon; KO Col = KO colon (**F and G**). Chromatograms of samples from the upper (**F**) and lower (**G**) SI of groups of WT and MUC2KO mice.

DOI: https://doi.org/10.7554/eLife.40387.011

The following figure supplement is available for figure 5:

**Figure supplement 1.** Ex vivo aggregation in 0.45 μm-filtered luminal fluid from the small intestines (SI) of wild-type (WT) and MUC2 knockout (MUC2KO) mice.

DOI: https://doi.org/10.7554/eLife.40387.012

summarized in *Tables 2,3* for both the upper and lower SI of MUC2KO and WT mice. We find that these estimates suggest there are some differences in the polymeric composition of the SI of these two groups.

To test whether these measured differences in polymeric composition are reflected in differences in aggregation, we looked at aggregation in the 0.45-µm-filtered samples. We found that the undiluted samples from both groups displayed aggregation (*Figure 5—figure supplement 1A–B*). We then created serial dilutions of the samples and found different aggregation thresholds for the samples (*Figure 5—figure supplement 1C–D*). These results further confirm our conclusion that although MUC2 may play a role in particle aggregation, it is not required for aggregation to occur.

## Immunoglobulins may play a role in aggregation, but are not required for aggregation to occur

It has also been demonstrated that immunoglobulins can bind to bacteria and induce them to aggregate (*Doe, 1989*; *Peterson et al., 2007*; *Levinson et al., 2015*; *Hendrickx et al., 2015*; *Endt et al., 2010*; *Bunker et al., 2017*; *Moor et al., 2017*; *Mantis et al., 2011*; *Donaldson et al., 2018*). We therefore wanted to test the hypothesis that immunoglobulins drive the aggregation of PEG-coated particles in the SI. To do this, we compared the aggregation of our PEG-coated particles in samples from groups of mutant mice that do not produce immunoglobulins (Rag1KO), to samples from groups of WT mice. Again, to carefully preserve the native composition of the SI fluid, we used a protease-inhibitor cocktail when collecting the samples (see Materials and methods). Because Rag1KO mice are immunocompromised, they need be fed an autoclaved chow diet. To control any potential differences in diet, both the Rag1KO and WT mice were fed an autoclaved chow diet for 48 hr before samples were collected.

The mice were confirmed to be Rag1KO via genotyping and Western blot (*Figure 6E*). According to the literature, IgA is abundant in the SI (*Murphy et al., 2004*). As expected, we saw a signal for IgA in the upper and lower SI of WT mice. We also tested for less abundant immunoglobulins such as IgG and IgM (*Figure 6—figure supplements 1* and *2*, respectively), but did not detect their presence in the luminal contents of either WT or KO mice.

We observed aggregation in 30-µm-filtered samples from Rag1KO and WT mice (*Figure 6A and B*). To test the strength of the aggregation effect in the different samples, we serially diluted the samples and compared the volume-weighted average aggregate sizes at each dilution (*Figure 6C and D*). We found differences in the amount of aggregation between the Rag1KO and WT samples at different dilutions, suggesting that although immunoglobulins are not required for aggregation to occur, they could play a role in the aggregation of PEG-coated particles.

We next wanted to test differences in the MW distribution of the polymers found in these samples, so we 0.45-µm-filtered our samples and analyzed them by GPC (see Materials and methods). The chromatograms from the refractometer (*Figure 6F and G*) suggested that the Rag1KO and WT samples were visually similar. We again made estimates of the physical parameters of the polymers in these samples (summarized in *Tables 4,5*). These estimates suggest that there are some differences in the polymeric composition of the SI of these two groups of mice.

**Table 2** Estimates of physical parameters of polymers from gel permeation chromatography for liquid fractions from the upper small intestine of MUC2 knockout (MUC2KO) and wild-type (WT) mice.

| Retention volume (mL) | 11 to 16 | | 16 to 20 | | >20 | |
|---|---|---|---|---|---|---|
| Mouse type | WT | MUC2KO | WT | MUC2KO | WT | MUC2KO |
| $M_w$ (kDa) | 3,560 ± 410 | 5,420 ± 620 | 162 ± 20 | 147 ± 17 | 4.05 ± 0.46 | 2.96 ± 0.34 |
| $M_w/M_n$ | 1.36 | 1.59 | 2.16 | 2.43 | 3.59 | 10.9 |
| $R_h$ (nm) | 49.1 | 45.5 | 6.31 | 5.95 | 1.18 | 1.02 |
| Fract. Conc. (mg/mL) | 2.52 ± 0.29 | 1.18 ± 0.13 | 24.6 ± 2.8 | 21.9 ± 2.5 | 88.7 ± 10.1 | 86.0 ± 9.8 |

We calculated values with both dn/dc = 0.185 (for proteins) and dn/dc = 0.147 (pullulan). When the value varied with dn/dc, it is reported in the table as the mid-range values ± the absolute deviation between the two calculated values. $M_w$ = the weight-average molecular weight; $M_w/M_n$ = the dispersity; $R_h$ = hydrodynamic radius; Fract. Conc. = Concentration of a given molecular weight fraction.

DOI: https://doi.org/10.7554/eLife.40387.013

**Table 3** Estimates of physical parameters of polymers from gel permeation chromatography for liquid fractions from the lower small intestine of MUC2 knockout (MUC2KO) and wild-type (WT) mice

| Retention volume (mL) | 11 to 16 | | 16 to 20 | | >20 | |
|---|---|---|---|---|---|---|
| Mouse type | WT | MUC2KO | WT | MUC2KO | WT | MUC2KO |
| $M_w$ (kDa) | 4,730 ± 540 | 5,180 ± 590 | 219 ± 25 | 155 ± 18 | 13.7 ± 1.6 | 5.93 ± 0.68 |
| $M_w/M_n$ | 1.24 | 1.80 | 1.91 | 1.84 | 1.88 | 2.03 |
| $R_h$ (nm) | 57.0 | 49.2 | 8.45 | 7.58 | 1.89 | 1.35 |
| Fract. Conc. (mg/mL) | 3.42 ± 0.39 | 2.36 ± 0.27 | 23.0 ± 2.6 | 22.8 ± 2.6 | 54.8 ± 6.3 | 63.3 ± 7.2 |

We calculated values with both dn/dc = 0.185 (for proteins) and dn/dc = 0.147 (pullulan). When the value varied with dn/dc, it is reported in the table as the mid-range values ± the absolute deviation between the two calculated values. $M_w$ = the weight-average molecular weight; $M_w/M_n$ = the dispersity; $R_h$ = hydrodynamic radius; Fract. Conc. = Concentration of a given molecular weight fraction.

DOI: https://doi.org/10.7554/eLife.40387.014

To test whether these measured differences in polymeric composition correspond with differences in aggregation, we quantified aggregation in the 0.45-µm-filtered samples. We found that the undiluted samples for both groups displayed aggregation (*Figure 6—figure supplement 3A and B*). When we created serial dilutions of the samples we found that the levels of aggregation were similar (*Figure 6—figure supplement 3C and D*). Taken together, the results suggest that immunoglobulins may play some role in aggregation, but the presence of immunoglobulins are not required for aggregation to occur.

Interestingly, there are some differences in the levels of aggregation in WT mice fed the autoclaved diet compared with the standard chow diet. The two diets are nutritionally the same, only the processing is different. When samples from the WT mice in the MUC2KO experiments are compared with samples from the WT mice in the Rag1KO experiments are compared, it is apparent that, compared with WT mice fed the normal chow diet, samples from WT mice fed the autoclaved diet had (i) a lower average concentration of polymers and (ii) polymers of lower overall MW (see 'WT' samples in *Tables 1–4*). These observations suggested two hypotheses: (*Donaldson et al., 2016*) dietary polymers may play a role in aggregation and (*McGuckin et al., 2011*) aggregation may be controlled by changing the polymer composition of the diet. We tested these hypotheses next.

### Polymers in the diet control aggregation of PEG-coated particles in a manner consistent with depletion-type interactions

As described in *Figure 4*, the extent of aggregation can be controlled by altering the polymer size and concentration of the polymer solution. Furthermore, as pointed out above, SI fluid from mice fed autoclaved and non-autoclaved diets induced different levels of aggregation. We hypothesized that aggregation behavior would differ between mice fed polymers of different sizes—even if the polymers were composed of similar chemical monomers and were present at the same polymer mass concentration. We hypothesized that mice fed short polymers would exhibit less aggregation in the SI (i.e. short polymers reduce the strength of the effect because depletion attractions are reduced). We predicted that the converse would be true for long polymers (i.e. long polymers increase the strength of the effect because depletion attractions are increased).

We first identified two candidate dietary carbohydrate polymers; Fibersol-2, a 'resistant maltodextrin' composed of D-glucose monomers (*Kishimoto et al., 2013*; *Fibersol, 2018*), with a MW of ~3500 Da (see *Table 6*) and apple pectin, composed of D-galacturonic acid and D-galacturonic acid methyl ester monomers (*Dongowski et al., 2000*; *Thakur et al., 1997*), with a MW of ~230 kDa (*Table 6*). Before feeding mice these polymers, we first tested their effects on aggregation in vitro at various concentrations in buffer (*Figure 7A*). We found similar trends to the PEG solutions in *Figure 4*. Pectin at low (~0.05 to ~1 mg/mL) and very high mass concentrations showed little aggregation (~7 mg/mL) and showed the most aggregation at an intermediate concentration (~1.5 to ~3 mg/mL). Fibersol-2 did not induce much aggregation up to a mass concentration of ~240 mg/mL.

To test our hypothesis that we could use polymer size to control aggregation, we devised a simple experiment. One group of mice was fed a solution of Fibersol-2 and a second group was fed a solution of apple pectin for 24 hr. The mass concentrations of the fibers in the two solutions were

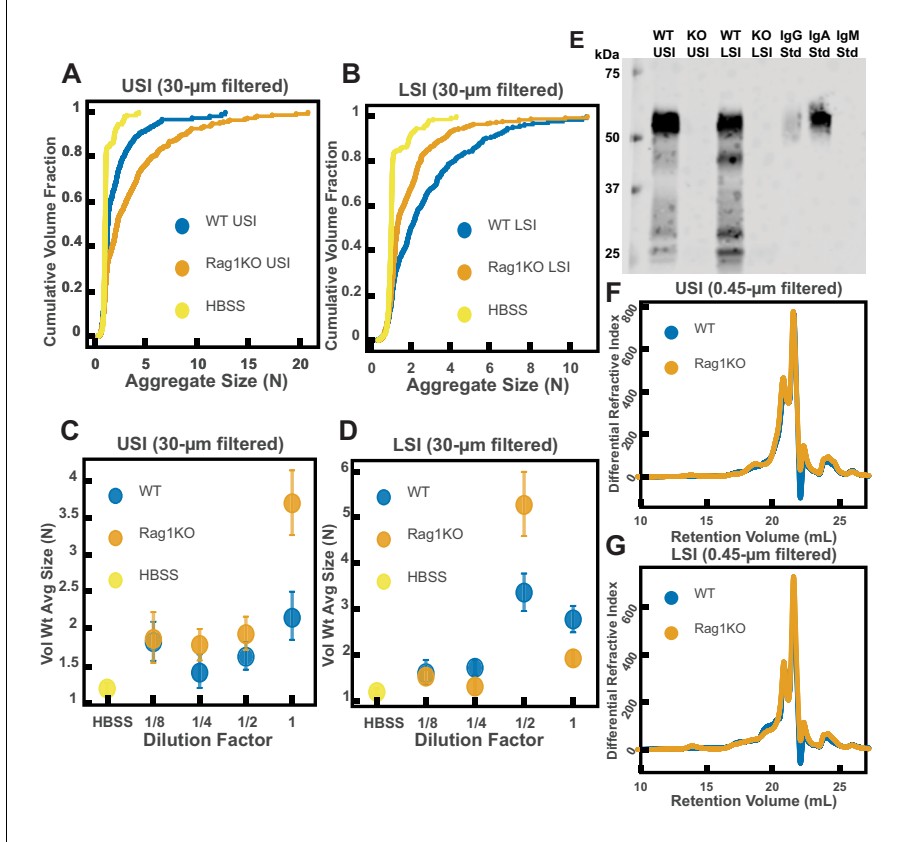

**Figure 6.** Quantification of the aggregation of particles in the small intestine (SI) in immunoglobulin-deficient (Rag1KO) and wild-type (WT) mice. (**A and B**) Volume-weighted empirical cumulative distribution functions (ECDFs) comparing aggregation of the particles in undiluted, 30 μm filtered samples from the upper (**A**) and lower (**B**) SI of two separate groups of wild-type (WT) and immunoglobulin-deficient (Rag1KO) mice to the control (particles suspended in HBSS). Plotted on the vertical axis is the cumulative volume fraction of the total number of particles in solution in an aggregate of a given size. Plotted on the horizontal axis are aggregate sizes in number of particles. (**C and D**). Volume-weighted average aggregate sizes (Vol Wt Avg Size) for serial dilutions of 30 μm filtered samples from the upper (**C**) and lower (**D**) SI of two separate groups of WT and Rag1KO mice. The dilution factor is plotted on the horizontal axis, where a dilution factor of 1 is undiluted, ½ is a two-fold dilution, and so on. The vertical error bars are 95% empirical bootstrap CI using the bootstrapping procedure described in Materials and methods. (**E**) Western blots of 30 μm filtered samples from the SI of WT and Rag1KO mice. WT USI = WT upper SI; KO USI = KO lower SI; WT LSI = WT lower SI; KO USI = KO upper SI. (**F and G**) Chromatograms of samples from the upper (**F**) and lower (**G**) SI of groups of WT and Rag1KO mice.

DOI: https://doi.org/10.7554/eLife.40387.015

The following figure supplements are available for figure 6:

**Figure supplement 1.** Western blots of 30 μm-filtered samples from the small intestine (SI) of wild-type (WT) and Rag1 knockout (Rag1KO) mice.

DOI: https://doi.org/10.7554/eLife.40387.016

**Figure supplement 2.** Western blots of 30 μm-filtered samples from the small intestine (SI) of wild-type (WT) and Rag1 knockout (Rag1KO) mice.

DOI: https://doi.org/10.7554/eLife.40387.017

**Figure supplement 3.** Ex vivo aggregation in 0.45-μm-filtered luminal fluid from the small intestines (SI) of wild-type (WT) and Rag1 knockout (Rag1KO) mice.

DOI: https://doi.org/10.7554/eLife.40387.018

matched at 2% w/v, and 5% w/v sucrose was added to each to ensure the mice consumed the solutions. Mesh-bottom cages were used to ensure that the mice did not re-ingest polymers from fecal matter via coprophagy. According to the literature, neither of these two polymers should be broken

**Table 4** Estimates of physical parameters of polymers from gel permeation chromatography for liquid fractions from the upper small intestine of immunoglobulin-deficient (Rag1KO) and wild-type (WT) mice.

| Retention volume (mL) | 11 to 16 | | 16 to 20 | | >20 | |
|---|---|---|---|---|---|---|
| Mouse type | WT | Rag1KO | WT | Rag1KO | WT | Rag1KO |
| $M_w$ (kDa) | 1,480 ± 170 | 2,140 ± 250 | 108 ± 12 | 74.2 ± 8.5 | 2.84 ± 0.32 | 1.91 ± 0.22 |
| $M_w/M_n$ | 1.09 | 1.14 | 2.62 | 2.42 | 1.59 | 1.54 |
| $R_h$ (nm) | 31.8 | 39.8 | 4.77 | 2.51 | 1.078 | 0.936 |
| Fract. Conc. (mg/mL) | 1.07 ± 0.12 | 1.13 ± 0.13 | 14.3 ± 1.6 | 13.9 ± 1.6 | 66.1 ± 7.6 | 70.5 ± 8.1 |

We calculated values with both dn/dc = 0.185 (for proteins) and dn/dc = 0.147 (pullulan). When the value varied with dn/dc, it is reported in the table as the mid-range value ± the absolute deviation between the two calculated values. $M_w$ = the weight-average molecular weight; $M_w/M_n$ = the dispersity; Rh = hydrodynamic radius; Fract. Conc. = Concentration of a given molecular weight fraction.

DOI: https://doi.org/10.7554/eLife.40387.019

down in the SI (*Fibersol, 2018*; *Holloway et al., 1983*; *Coenen et al., 2006*). As before, all samples were collected with a protease-inhibitor cocktail.

We created serial dilutions of the small intestinal luminal fluid and looked at the extent of aggregation in each sample. In the 30-µm-filtered samples from the upper SI we observed more aggregation in the pectin-fed mice compared with the Fibersol-2 fed mice (*Figure 7E*). For the undiluted 30-µm-filtered lower SI sample, the pectin-fed mice samples formed a gel-like material which we were unable to pipette and therefore could not use for aggregation experiments. This gelation is not too surprising considering that pectin can form a gel in certain contexts (*Thakur et al., 1997*; *Saha and Bhattacharya, 2010*). We were able to dilute this gel four-fold and then compare the aggregation in serial dilutions of the pectin-fed LSI to the Fibersol-2-fed LSI. We found, again, more aggregation in the pectin-fed mice than the Fibersol-2-fed mice (*Figure 7G*).

We again 0.45-µm-filtered these samples and ran them on GPC to test differences in the MW and size distributions of the polymers in these samples. The chromatograms from the refractometer (*Figure 7C and D*) suggest that there are differences in the polymeric distribution in the two groups of mice. *Figure 7B* shows chromatograms of just Fibersol-2 and pectin in buffer. We see that pectin elutes between 14–18 min, which is where we see an enhancement of the concentration of high-MW polymers in the samples from the SIs of the group fed pectin. We also see that Fibersol-2 elutes between 18–22 min, which is where we see an enhancement in the concentration of low-MW polymers in the samples from the SI of the group fed Fibersol-2. We again made estimates of the physical parameters of the polymers in these samples which are summarized in *Tables 7,8*. The estimates also suggest that there are differences in the polymeric composition of the SI of the two groups. Overall, the data from GPC suggests that the pectin-fed mice have more high-MW polymers than the Fibersol-2-fed mice. Low-MW polymers appear to be more abundant in Fibersol-2 fed mice compared with pectin-fed mice. We observed visually that the SI contents of the pectin-fed mice formed a gel and pectin is also known to self-associate to form a gel or aggregates in solution

**Table 5** Estimates of physical parameters of polymers from gel permeation chromatography for liquid fractions from the lower small intestine of immunoglobulin-deficient (Rag1KO) and wild-type (WT) mice.

| Retention volume (mL) | 11 to 16 | | 16 to 20 | | >20 | |
|---|---|---|---|---|---|---|
| Mouse type | WT | Rag1KO | WT | Rag1KO | WT | Rag1KO |
| $M_w$ (kDa) | 1,080 ± 120 | 2,490 ± 290 | 66.9 ± 7.7 | 91.6 ± 10.5 | 3.64 ± 0.42 | 3.72 ± 0.43 |
| $M_w/M_n$ | 1.18 | 1.05 | 1.71 | 1.98 | 2.09 | 1.98 |
| $R_h$ (nm) | 34.6 | 47.1 | 4.67 | 4.85 | 1.116 | 1.09 |
| Fract. Conc. (mg/mL) | 1.52 ± 0.17 | 1.89 ± 0.22 | 15.8 ± 1.8 | 14.1 ± 1.6 | 49.5 ± 5.7 | 55.1 ± 6.3 |

We calculated values with both dn/dc = 0.185 (for proteins) and dn/dc = 0.147 (pullulan). When the value varied with dn/dc, it is reported in the table as the mid-range values ± the absolute deviation between the two calculated values. $M_w$ = the weight-average molecular weight; $M_w/M_n$ = the dispersity; $R_h$ = hydrodynamic radius; Fract. Conc. = Concentration of a given molecular weight fraction.

DOI: https://doi.org/10.7554/eLife.40387.020

**Table 6** Gel permeation chromatography of Fibersol-2 and pectin in phosphate-buffered saline

| Sample | Fibersol-2 | Pectin |
|---|---|---|
| $M_w$ (kDa) | 3.48 | 232 |
| $M_w/M_n$ | 10.5 | 1.97 |
| $R_h$ (nm) | 1.24 | 25.4 |

Both fiber types were analyzed with dn/dc = 0.147 for polysaccharides. $M_w$ = weight-average molecular weight; $M_w/M_n$ = the dispersity; $R_h$ = hydrodynamic radius

DOI: https://doi.org/10.7554/eLife.40387.022

(*Thakur et al., 1997*; *Saha and Bhattacharya, 2010*). We note, therefore that by 0.45-µm-filtering these samples we may be removing these structures and decreasing the concentration of pectin in our samples.

To test that these measured differences in polymeric composition are reflected in differences in aggregation, we tested aggregation in the 0.45-µm-filtered samples. We found that in both the upper and lower SI samples, the samples from the pectin-fed group showed more aggregation than the samples from the group fed Fibersol-2 (*Figure 7F and H*). When we created serial dilutions of these samples, we found that the samples from the mice fed Fibersol-2 showed almost no aggregation at any concentration whereas the samples from pectin-fed mice showed aggregation. We also observed that we needed to dilute the 30-µm-filtered samples more to achieve the greatest extent of aggregation (*Figure 7E and G*). We speculate that this shift in the aggregation behavior between the 30-µm-filtered and 0.45-µm-filtered samples is due to some of the polymers being lost when 0.45-µm-filtering the samples as a result of the aforementioned self-association of pectin.

These data taken together lead us to conclude that polymers in the diet can be used to control the aggregation of PEG-coated particles. This data further suggests that feeding higher MW polymers at the same mass concentration as lower MW polymers leads to an enhancement in aggregation. Due to the high polydispersity and complex chemical composition of SI luminal fluid as measured by GPC, it is unfeasible to apply the same theoretical analysis as was done in *Figure 4* to these data. We can, however, note that visually the behavior is qualitatively consistent with the depletion-type interactions found in simple PEG solutions in *Figure 4*.

## Discussion

This work shows that even PEG-coated particles, which have minimal biochemical interactions, form aggregates in the luminal fluid of the SI. It reveals a previously unknown way in which dietary polymers can impact, and be used to control, the structure of particles in the SI. We speculate that this phenomenon may play a role in the aggregation of other particles in the SI such as microbes, viruses, nanoparticles for drug delivery, and food granules. For these types of particles, other factors will also inevitably affect the formation of aggregates (e.g. interactions with mucins and immunoglobulins); thus, it will be important to explore the interplay among all these factors. Another important next step is to investigate how mixing in the SI and the co-aggregation of different types of particles may affect aggregation. We speculate that the aggregation of particles in the SI could also have functional consequences, such as promoting colonization by microbes, affecting infection by pathogens, and altering clearance of microbes (*McGuckin et al., 2011*; *Millet et al., 2014*; *Lukic et al., 2014*; *Del Re et al., 1998*; *Tzipori et al., 1992*; *Howe et al., 2014*). Aggregation will also need to be considered when designing nanoparticles for drug delivery (*Maisel et al., 2015a*; *Goldberg and Gomez-Orellana, 2003*).

We found that MUC2 and immunoglobulins, which have been found to aggregate microbes both in vivo and in vitro (*Puri et al., 2015*; *Laux et al., 1986*; *Sajjan and Forstner, 1990*; *Wanke et al., 1990*; *Sun et al., 2007*; *Doe, 1989*; *Peterson et al., 2007*; *Levinson et al., 2015*; *Hendrickx et al., 2015*; *Endt et al., 2010*; *Bunker et al., 2017*; *Moor et al., 2017*; *Mantis et al., 2011*; *Donaldson et al., 2018* ), are not required for the aggregation of PEG-coated particles. Instead, we found that by feeding mice dietary polymers with similar chemistry but very different sizes we could tune the extent of aggregation in the SI. These polymers (pectin and Fibersol-2) are forms of fiber commonly found in the human diet. We found that feeding long polymers induced aggregation,

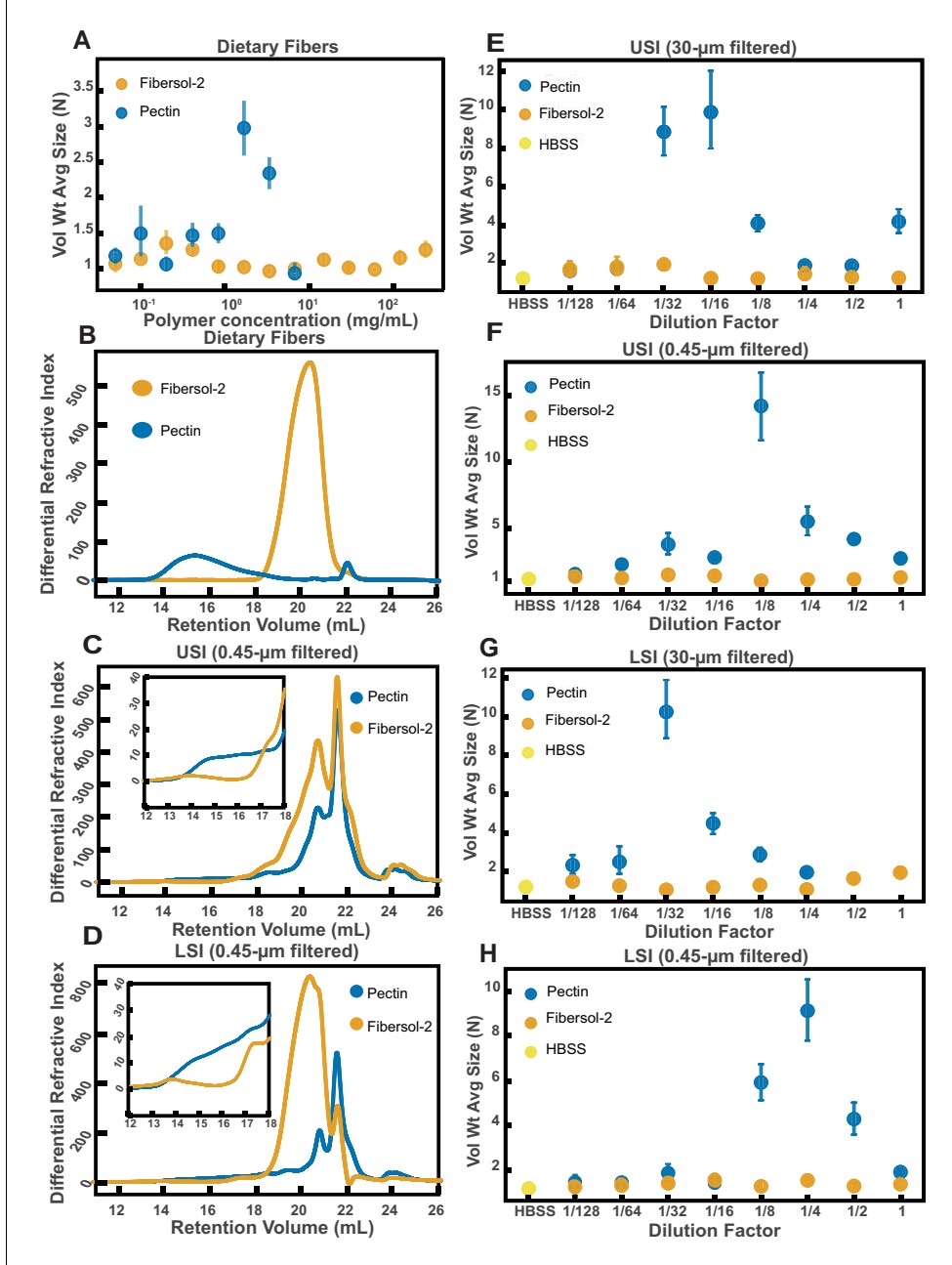

**Figure 7.** Quantification of aggregation of PEG-coated particles in the small intestine (SI) of mice fed different polymers from dietary fiber. (**A**) Volume-weighted average aggregate sizes (Vol Wt Avg Size) for serial dilutions of apple pectin and Fibersol-2. Volume-weighted average sizes are plotted on the vertical axis in terms of number of particles per aggregate (N) against polymer concentration (mg/mL). The vertical error bars are 95% empirical bootstrap CI using the bootstrapping procedure described in Materials and methods. (**B**) Chromatograms of apple pectin and Fibersol-2 in buffer. (**C and D**) Chromatograms of samples from the upper (**E**) and lower (**F**) SI of two separate groups of mice (fed pectin or Fibersol-2). (**E-H**) Volume-weighted average aggregate sizes (Vol Wt Avg Size) for serial dilutions of 30-μm-filtered samples from the upper (**E**) and lower (**G**) SI of two separate groups of mice (fed pectin or Fibersol-2) to the control (particles suspended in HBSS). (**F and H**) Serial dilutions of 0.45-μm-filtered samples from the same groups. The dilution factor is plotted on the horizontal axis, where a dilution factor of 1 is undiluted, and ½ is a two-fold dilution. The vertical error bars are 95% empirical bootstrap CI using the bootstrapping procedure described in Materials and methods.

DOI: https://doi.org/10.7554/eLife.40387.021

**Table 7** Estimates of physical parameters of polymers from gel permeation chromatography for liquid fractions from upper small intestine of pectin and Fibersol-2 fed mice.

| Retention volume (mL) | 11 to 16 | | 16 to 20 | | >20 | |
|---|---|---|---|---|---|---|
| Mouse type | Pectin | Fibersol-2 | Pectin | Fibersol-2 | Pectin | Fibersol-2 |
| $M_w$ (kDa) | 267 ± 31 | 686 ± 79 | 40.0 ± 4.5 | 35.3 ± 4.0 | 1.39 ± 0.16 | 1.67 ± 0.19 |
| $M_w/M_n$ | 1.50 | 1.08 | 2.15 | 2.64 | 2.45 | 1.48 |
| $R_h$ (nm) | 31.8 | N/C** | 5.52 | 2.88 | 0.819 | N/C** |
| Fract. Conc. (mg/mL) | 1.62 ± 0.19 | 0.516 ± 0.059 | 9.00 ± 1.03 | 23.3 ± 2.7 | 53.7 ± 6.1 | 77.0 ± 8.8 |

We calculated values with both dn/dc = 0.185 (for proteins) and dn/dc = 0.147 (pullulan). When the value varied with dn/dc, it is reported in the table as the mid-range values ± the absolute deviation between the two calculated values. $M_w$ = the weight-average molecular weight; $M_w/M_n$ = the dispersity; $R_h$= hydrodynamic radius; Fract. Conc. = Concentration of a given molecular weight fraction. N/C** denotes values for which the concentration was too low to calculate.
DOI: https://doi.org/10.7554/eLife.40387.023

whereas short polymers did not. More work needs to be done to understand the underlying mechanism, but surprisingly the observed aggregation behavior in the SI luminal fluid from mice fed dietary polymers of different sizes is qualitatively consistent with the aggregation behavior in simple PEG solutions, where aggregation is driven by depletion interactions. Overall, this suggests a simple dietary method for controlling aggregation in the gut. It will be important to extend this work to microbes and other particles commonly found in the gut and to measure the relative contributions of polymer-driven aggregation and chemical-driven aggregation. We note that mucins and immunoglobulins are polymers that can also self-associate into structures of very high MW (*Ambort et al., 2012*; *Grey et al., 1971*; *Kerr, 1990*), suggesting that they could cause aggregation via both physical and chemical mechanisms. Interestingly, during the review of this manuscript, a study was published with in vitro work done using model buffer solutions of mucins, DNA, and other biopolymers further implying that aggregation of bacteria by host-polymers can be depletion-mediated (*Secor et al., 2018*). In vivo, it will also be important to consider the effects of flow, as it has been shown that flow in non-Newtonian fluids can induce particle aggregation (*Highgate, 1966*; *Michele et al., 1977*; *Kim and Helgeson, 2016*). In particular, studies have suggested that the combination of flow and polymer elasticity can lead to aggregation (*Highgate and Whorlow, 1970*) and that shear thinning viscosity can influence aggregation as well (*Snijkers et al., 2013*). In our work, we neglected flow effects for simplicity and thus our findings are most applicable to the initial formation of aggregates before aggregation is influenced by mechanical forces due to peristaltic mixing and the transit of food. A rudimentary estimate of the Weissenberg number (see Materials and methods), which weighs the contributions of elastic and viscous forces, yields $\mathrm{Wi} \sim 0.3 \text{ to } 10$, suggesting that elasticity-induced effects may play a role in the SI and will be an important direction to pursue in follow-up studies. If flow-induced clustering does occur in vivo, the literature suggests it would aid in the process, perhaps enhancing particle aggregation.

**Table 8** Estimates of physical parameters of polymers from gel permeation chromatography for liquid fractions from lower small intestine of pectin and Fibersol-2-fed mice.

| Retention volume (mL) | 11 to 16 | | 16 to 20 | | >20 | |
|---|---|---|---|---|---|---|
| Mouse type | Pectin | Fibersol-2 | Pectin | Fibersol-2 | Pectin | Fibersol-2 |
| $M_w$ (kDa) | 282 ± 32 | 1680 ± 190 | 30.2 ± 3.5 | 18.8 ± 2.2 | 1.12 ± 0.13 | 2.32 ± 0.27 |
| $M_w/M_n$ | 7.37 | 1.64 | 1.70 | 2.78 | 2.89 | 1.14 |
| $R_h$ (nm) | 29.0 | 26.4 | 5.28 | 2.16 | 0.724 | 1.06 |
| Fract. Conc. (mg/mL) | 2.48 ± 0.28 | 0.839 ± 0.096 | 9.43 ± 1.1 | 53.6 ± 6.1 | 42.7 ± 4.9 | 88.3 ± 10.1 |

We calculated values with both dn/dc = 0.185 (for proteins) and dn/dc = 0.147 (pullulan). When the value varied with dn/dc, it is reported in the table as the mid-range values ± the absolute deviation between the two calculated values. $M_w$ = the weight-average molecular weight; $M_w/M_n$ = the dispersity; $R_h$ = hydrodynamic radius; Fract. Conc. = concentration of a given molecular weight fraction.
DOI: https://doi.org/10.7554/eLife.40387.024

We note that current dietary guidelines do not differentiate between fibers of low and high MW (*USDA, 2007*; *USDA, 2015*). Our work implies that the MW of fiber, and the subsequent degradation of a high-MW fiber into a low-MW component (*Datta et al., 2016*), which we have discussed previously in the context of mucus compression, is important in defining the physicochemical environment of the gut. Further studies will be required to understand the effects of industrial food processing on MW of the dietary polymers present in foods, and which processing methods preserve or produce high-MW polymers that impact mucus compression (*Datta et al., 2016*) and particle aggregation in the gut.

# Materials and methods

## Key resources table

| Reagent type (species) or resource | Designation | Source or reference | Identifiers | Additional information |
|---|---|---|---|---|
| MUC2KO, C57BL/6 mice (female) | MUC2KO | Eugene Chang Lab (University of Chicago) provided initial breeding pairs which were provided to them from Leonard H. Augenlicht at the Department of Oncology of Albert Einstein Cancer Center | | Genotyping was performed by Transnetyx Inc.; Western blot was done to confirm lack of MUC2 (See *Figure 5E*) |
| Rag1KO, C57BL/6 mice (male) | Rag1KO | Provided by Mazmanian Lab at Caltech | RRID:IMSR_JAX:002216 | Western blot was done to confirm lack of IgA as explained in the text (See *Figure 6E*) |
| C57BL/6 mice (all male except for WT controls in MUC2KO experiments in *Figure 5*) | WT | The Jackson Laboratory | RRID:IMSR_JAX:000664 | |
| Antibody | MUC2 polyclonal antibody (rabbit host) | Biomatik | Cat No: CAU27315 | |
| Antibody | Li-Cor IRDye 800 CW Goat Anti-Rabbit IgG | Li-Cor | P/N 925–32211; RRID:AB_2651127 | |
| Antibody | Li-Cor IRDye 800 CW Goat Anti-Mouse IgG | Li-Cor | P/N 925–32210; RRID:AB_2687825 | |
| Antibody | Li-Cor IRDye 800 CW Goat Anti-Mouse IgM | Li-Cor | P/N 925–32280 | |
| Antibody | Goat Anti-Mouse IgA-unlabeled | Southern Biotech | Cat No: 1040–01 | |
| Antibody | Li-Cor IRDye 800 CW Donkey Anti-Goat IgG | Li-Cor | P/N 925–32214; RRID:AB_2687553 | |
| Chemical compound, drug | apple pectin | Solgar Inc. | 'Apple pectin powder'; SOLGB70120 00B | |
| Chemical compound, drug | Fibersol-2 | Archer Daniels Midland/ Matsutani LLC | Product code: 013100, Lot #: CY4P28540 | |

*Continued on next page*

*Continued*

| Reagent type (species) or resource | Designation | Source or reference | Identifiers | Additional information |
|---|---|---|---|---|
| Chemical compound, drug | USP grade sucrose | Sigma-Aldrich | | |
| Chemical compound, drug | Protease inhibitor cocktail | Roche cOmplete, Mini, EDTA-free Protease-Inhibitor cocktail, Roche | | |
| Chemical compound, drug | PEG 100 kDa | Dow | POLYOX WSR N-10 | |
| Chemical compound, drug | PEG 1 MDa | Dow | POLYOX WSR N-12K | |
| Chemical compound, drug | PEG 3350 | Bayer | MiraLAX | |
| Chemical compound, drug | Hanks' Balanced Salt Solution (without calcium, magnesium, phenol red) | GE Healthcare Life Sciences | Product code: SH30588.02 | |
| Software, algorithm | 3D aggregate analysis pipeline | This paper; source code available through Dryad | | Description in Materials and methods; source code provided on Dryad |
| Other | mesh-bottom (or wire-bottom) floors | Lab Products, Inc. | P/N: 75016 | |
| Other | 1 μm diameter PEG 5 kDa-coated polysytrene beads | This paper | | Description of synthesis in Materials and methods |
| Other | 1 μm diameter PEG 5 kDa-coated polysytrene beads with PEG 1 kDa 'back-filling' | This paper | | Description of synthesis in Materials and methods |
| Other | standard chow diet | PicoLab | PicoLab Rodent Diet 20; Product #5053 | |
| Other | autoclaved chow diet | PicoLab | Laboratory Autoclavable Rodent Diet 5010 | |

## Details of animals used

All mice were male or female specific pathogen free (SPF) C57BL/6 mice between 8–16 weeks old. Mice on a standard, solid chow diet were given food and water *ad libitum.* Immunoglobulin-deficient (Rag1KO) mice were maintained on an autoclaved chow diet due to their immunocompromised status. The control group of WT mice used as a comparison to this group was maintained on the same autoclaved chow diet for 48 hr before euthanasia. Genotyping of MUC2 deficient (MUC2KO) and Rag1KO mice was done by Transnetyx (Transnetyx, Inc., Cordova, TN, USA). Mice given only apple pectin (Solgar, Inc., Leonia, NJ, USA) with sucrose (USP grade, Sigma-Aldrich, St. Louis, MO, USA) or Fibersol-2 (Archer Daniels Midland/Matsutani LLC, Chicago, IL, USA) with sucrose were first raised on a standard chow diet and given water *ad libitum,* then were maintained on a restricted diet consisting of only 2% apple pectin +5% sucrose or 2% Fibersol-2 +5% sucrose for 24 hr. For those 24 hr, these mice were kept on mesh-bottom cages to prevent the re-ingestion of polymers from the standard chow diet via coprophagy. The MUC2KO colony was raised and maintained by the

Ismagilov Lab. The Rag1KO mice were provided by the Mazmanian lab (Caltech). All other mice were from Jackson Labs (The Jackson Laboratory, Bar Harbor, ME, USA). All animal experiments were approved by the California Institute of Technology (Caltech) Institutional Animal Care and Use Committee (IACUC) and the U.S. Army's Animal Care and Use Review Office (ACURO). Mice were euthanized via $CO_2$ inhalation as approved by the Caltech IACUC in accordance with the American Veterinary Medical Association Guidelines on Euthanasia (*Leary et al., 2013*).

## Oral administration of particles

Particles were gavaged at a concentration of 0.1–2% w/v in either 1x HBSS or 1x PBS. We used small fluid volumes (50 µL) to minimize volume-related artifacts (*Maisel et al., 2015a*). We chose buffers isotonic to the SI because it has been shown that the isotonicity of the delivery medium can greatly affect the in vivo particle distribution (*Maisel et al., 2015b*). In some experiments, animals were food-restricted for 4 hr prior to administration of particles. It has been previously demonstrated though that food-restriction has minimal effects on the in vivo distribution of PEG-coated particles (*Maisel et al., 2015a*). In all experiments animals were euthanized 3 hr after administration of particles.

## Fluorescent scanner experiments

Gastrointestinal tracts (GIT) were excised and laid out flat on petri dishes on ice. Drops of saline were then placed around the GIT and the petri dishes were sealed with parafilm. Samples were then immediately brought to the fluorescent laser scanner (Typhoon FLA 9000) for imaging. Samples were scanned with an excitation wavelength of 473 nm and a 530 nm bandpass filter.

## Imaging of luminal contents from mice orally administered particles

Immediately after euthanization the small intestines of the mice were excised and divided into an upper and lower section. The luminal contents were collected by gently squeezing the intestines with tweezers. They were placed directly onto a glass slide and encircled by a ring of vacuum grease that did not touch the contents. A coverslip was then immediately placed on top to create an air-tight chamber. Samples were kept on ice during the collection process. The samples were then immediately taken for imaging. All imaging was performed using a Zeiss LSM 800 or a Leica DMI6000, using either bright-field microscopy, epifluorescence microscopy (GFP, L5 Nomarski prism), confocal fluorescence microscopy (488 nm excitation and 490–540 nm detection), or confocal reflectance microscopy (561 nm excitation and 540–700 nm detection).

## Collection of intestinal luminal fluid

Immediately after euthanasia, the SI of each mouse was excised and divided into an upper and lower section. If luminal fluid was collected from the colon, then the colon was also excised. The luminal contents were then collected from each section in separate tubes and kept on ice. The luminal contents from an individual mouse were insufficient in volume to perform all the required analyses (i.e. ex vivo aggregation, GPC, and sometimes Western blot), so contents were pooled from a group of three mice of the same age that were co-housed. These pooled samples, kept divided by section, were then spun down at 17 kG at 4°C for 1 hr to separate the liquid and solid portions of the contents. The supernatant of each sample was collected and then placed on 30 µm filters (Pierce Spin Columns – Snap Cap, Thermo Fisher Scientific, Waltham, MA, USA) and spun down at 17 kG at 4°C for 1 hr. Part of the filtrates of each sample were then collected, divided into aliquots, and frozen at −20°C for future experiments. The remaining portion of the filtrates was then taken and placed on 0.45 µm centrifugal filters (Corning Costar Spin-X centrifuge tube filters; cellulose acetate membrane, pore size 0.45 µm, sterile) and spun down at 5 kG at 4°C for 1 hr. For experiments in which a protease-inhibitor cocktail (Roche cOmplete, Mini, EDTA-free Protease-Inhibitor Cocktail, Roche, Indianapolis, IN, USA) was used, a 100x concentrated stock solution was prepared in HBSS (without calcium, magnesium, and phenol red; GE Healthcare Life Sciences, Marlborough, MA, USA). The same procedure as detailed above were followed for the collection of luminal fluid, except immediately after the luminal contents were brought back from the animal facility on ice, 10 µL of the 100x protease-inhibitor cocktail was added to each tube. The mixtures were then vortexed briefly to mix. The contents were then spun down at 17 kG at 4°C as described above to separate the solid from

liquid contents. The liquid fraction collected from each group before 30 and 0.45 μm filtration was usually ~200–300 mL, so the additional 10 μL of protease-inhibitor cocktail only diluted the samples by ~5% at most.

## Ex vivo and in vitro aggregation assays

We took 1 μm diameter PEG 5 kDa-coated polystyrene beads (with PEG 1 kDa 'back-filling') and suspended them at 10 mg/mL in deionized water. Before use, they were vortexed to re-suspend in solution and then sonicated for 1 min. The particle solution was then added to the polymer solution or small intestinal luminal fluid at a ratio of 1:10. After addition of particles, the mixture was vortexed for 10 s. Then, 2 μL of the mixture was then immediately pipetted into an imaging chamber created with a SecureSeal imaging spacer (0.12 mm depth and 9 mm diameter, Electron Microscopy Sciences, Hatfield, PA, USA) and a glass slide. The top of the imaging chamber was immediately sealed with a #1.5 coverslip. The samples were then imaged approximately 10 min later. In PEG solution experiments and serial dilution experiments, HBSS (without calcium, magnesium, phenol red; GE Healthcare Life Sciences) was used to dilute.

In the 1 MDa PEG experiments conducted in phosphate buffered saline (PBS) with pH = 6 (*Figure 4—figure supplement 2*) the PBS solution was initially prepared with 138 mM sodium chloride, 7.5 mM monosodium phosphate dihydrate, 1.1 mM disodium phosphate heptahydrate, and deionized (DI) water (Milli-Q). The sodium chloride was added to ensure that the ionic strength matched that of HBSS. The pH was then measured using an Orion 2-Star Benchtop pH Meter (Thermo Scientific) with an Orion 9110DJWP Double Junction pH electrode (Thermo Fisher Scientific) after first calibrating the instrument using the following reference standard buffers: pH = 10 (VWR BDH5078-500 mL), pH = 7 (VWR BDH5046-500 mL), and pH = 4 (VWR BDH5024-500 mL). The pH of the solution was then adjusted to pH = 6 using 1 M NaOH in DI water.

## Microscopy for ex vivo and in vitro aggregation assays

All imaging was performed using a Zeiss LSM 800, using confocal fluorescence microscopy (488 nm excitation, detection at 490–540 nm). We collected 3D stacks which were 200 × 200 × 40 μm in volume. 3D renders of aggregates were created using Imaris software from Bitplane, an Oxford Instruments Company.

## Imaging analysis

All image analysis was done in FIJI (ImageJ 2.0.0) using an ImageJ macro written using the ImageJ macro scripting language. These macros are available in Dryad. Z-stacks were saved as 16 bit .czi files and were subsequently loaded into FIJI. Each z-stack extended ~40 μm deep into each sample in the z-direction and was composed of 113 slices. As a result of the depth of the stacks in the z-direction, we observed a significant drop-off in measured aggregate fluorescence between the first slice and the last slice, likely due to scattering from the intestinal fluid and the particles themselves. To ensure that aggregates throughout a given stack had a similar brightness, which is important for the 3D Object Counter plugin, the median pixel intensity for aggregates in every slice was set as the maximum pixel intensity value for every slice. To achieve this, first the 10th slice and the 10th to last slice of the z-stack were selected and thresholded using the Otsu method (*Otsu, 1979*), creating a binary image of the aggregates in the two slices. The binary images were used as masks to measure the median pixel intensity of each aggregate in the two slices as well as the mean and max pixel intensity values for the background of both images. The drop-off in intensity was assumed to be approximately linear, so the median pixel intensity for aggregates in each slice was determined by interpolating between the median aggregate pixel intensity values from the 10th slice and 10th to last slice. The minimum pixel intensity value for each slice was determined by adding 1/3 of the mean background pixel intensity to 2/3 of the maximum background pixel intensity for the 10th and 10th to last slices (this was necessary to deal with the challenge determining background pixel intensities) and then interpolating to calculate the minimum for all other slices. The process of intentionally introducing image clipping in the z-stacks was justified by the manner in which aggregates were identified; aggregates were first measured by total volume instead of by particle count, thus being able to discern individual particles inside of each aggregate was unnecessary.

The 3D Objects Counter plugin in FIJI was used to measure various parameters, including the volume of each aggregate. The plugin initially thresholds all slices in a stack using a single thresholding value, which requires objects in every slice of a stack to be roughly the same intensity (hence, the thresholding procedure described previously). The plugin takes the resulting now-binary z-stack and determines the number of voxels occupied by each aggregate and converts voxel volume to metric volume using metadata in each .czi file. A second macro was used to determine the average size of a singlet (single particle) for each z-stack. In this macro, we identified 10 singlets by visually inspecting the sample to determine the average size of a singlet. This was then used to normalize differences in measured aggregate volume between samples by converting to a particle count per aggregate. This normalization step was necessary due to variations in the average measured singlet size between samples. It also helped account for any differences in the thresholding procedure from sample to sample.

The accuracy of this method for determining aggregate sizes was validated by comparing empirical cumulative distribution functions (ECDFs) of the cross-sectional area of the aggregates in a given z-stack determined by the ImageJ macro to ECDFs generated by visually inspecting the samples to measure the cross-sectional areas of aggregates. This comparison was done for at least three separate z-stacks. ImageJ macros will be made available upon request.

## Quantification of aggregate sizes

The sizes of aggregates in solution were quantified in two ways. One was by comparing the volume-weighted empirical cumulative distribution functions (ECDFs) of the aggregate sizes of each sample to each other. The volume-weighted ECDF, $F$, as follows (*Bois, 2017*):

$$\hat{F}(N) = \frac{1}{\sum N_i} \sum_{i=1}^{n} I(N_i \leq N) \tag{6}$$

$$I(N_i \leq N) = \begin{cases} N_i \ if \ N_i \leq N \\ 0 \ if \ N_i > N \end{cases} \tag{7}$$

Where $N_i$ is the number of particles per aggregate and $n$ is the total number of aggregates in solutions (where single particles also count as aggregates).

The other way in which the extent of aggregation was quantified was by creating bootstrap replicates of the ECDFs of the aggregate distributions of each sample and computing the volume-weighted average aggregate size ($\langle N \rangle$; given in number of particles per aggregate) for each bootstrap replicate. The volume-weighted average aggregate size is given by the following equation in units of 'number of particles per aggregate':

$$\langle N \rangle = \frac{\sum_{i=1}^{n} N_i^2}{\sum_{i=1}^{n} N_i} \tag{8}$$

This allowed us to calculate 95% empirical bootstrap CI on the volume-weighted average aggregate size. We generated 10,000 bootstrap replicates from the original ECDF of each sample to generate these. The advantage of this approach is that we do not need to assume anything about the underlying probability distribution; it is non-parametric (*Bois, 2017*). The original ECDFs, from which the replicates were generated, each contained at least 300 aggregates, in many cases containing ~1000 or more aggregates. The codes used for the analyses (volume-weighted ECDFs and 95% empirical bootstrap CIs) were written in Python 3.6.4 and are available on Dryad.

## Filtration with MW cut-off filters

Small intestinal luminal fluid was collected and 0.45-μm-filtered as described in 'Collection of Luminal Fluid'. It was then divided up and placed on MWCO filters (Pierce Protein Concentrators, Thermo Fisher Scientific) with the following MWCOs: 100 kDa, 30 kDa, and 3 kDa. The samples were then centrifuged at 15 kG at 4°C for 2 hr, checking every 15 min for the first hour if additional volume had flowed through. After the eluent from each was collected, they were diluted back to their original volumes with HBSS.

## pH measurements of luminal fluid

Pooled samples of luminal fluid were collected from each section (stomach, upper small intestine, lower small intestine, cecum, and colon) and 30-µm-filtered as described in 'Collection of Luminal fluid' (with use of the same protease inhibitor cocktail). Samples were collected from two separate groups of 2-month-old B6 male mice on a standard chow diet. Each group had three mice. Because there was only ~25 µL of luminal fluid from the colons of each group we did not 30 µm-filter the colonic fluid as there was concern all the fluid would be retained by the filter. The colonic contents were simply spun down at 17 kG at 4°C for 1 hr to separate the liquid and solid portions of the contents. Then the supernatant (luminal fluid) was collected. Measurements were done using an Orion 2-Star Benchtop pH Meter. The instrument was first calibrated with three reference standard buffers: pH = 10 (VWR BDH5078-500 mL), pH = 7 (VWR BDH5046-500 mL), and pH = 4 (VWR BDH5024-500 mL). Measurements were conducted at T = 25°C. There was at least 100 µL of sample from each section except for the stomach sample from one group of mice and from colon samples from both groups. Measurements were conducted with both a standard pH electrode (Orion 9110DJWP Double Junction pH Electrode) and a micro pH electrode (Orion 9810BN Micro pH Electrode, Thermo Fisher Scientific). This was done because the standard electrode is only accurate for samples with volumes of 200 µL whereas the micro electrode is accurate for samples as small as 0.5 µL in volume. The results are consistent with other results for rodents (*Ward and Coates, 1987*; *Smith, 1965*) with the exception of a study conducted with mice of a different gender, strain, and fed an 18% protein diet (*McConnell et al., 2008*).

For the pH measurement of HBSS, the pH was measured with both the standard and micro pH electrodes, and three technical replicates were done with each probe. The value for the pH reported in the main text is the average of all six measurements.

## Estimation of coverage and length of grafted PEG layer

Based on our NMR measurements (see section 'NMR of PEG-coated particles with "backfill"') the grafting density ($\Gamma$) of the PEG polymer on our PEG 5 kDa-coated particles with PEG 1 kDa backfill should be approximately: $\Gamma = 0.48$ chains/nm$^2$ (to estimate this we assume that all of the PEG on the surface is PEG 5 kDa). One can estimate the grafting density at which the grafted chains transition from separate coils to overlapping coils or the brush regime by calculating the grafting density at which coils would just begin to overlap (*de Gennes, 1980*). This can be estimated as:

$$\Gamma^* \sim \frac{1}{\pi R_g^2} \qquad (9)$$

Where $R_g$ is the radius of gyration of the grafted polymer. Using literature measurements of the hydrodynamic radius of PEG 5 kDa and the Kirkwood-Riseman relation, this can be estimated as $R_g \sim 3.45\ \mathrm{nm}$. We therefore estimate that $\frac{\Gamma}{\Gamma^*} \sim 5$, meaning that the grafting density is such that the polymer coils on the surface should be overlapping and within the brush regime. To estimate the length and average volume fraction of the layer, we therefore made the assumption that the grafted polymer layer behaved as a brush and used the Alexander-deGennes brush approximation (*Rubinstein and Colby, 2003*; *Israelachvili, 2011*). This theory was originally developed for high-MW polymer coils, but has also been found, surprisingly, to quantitatively capture forces for grafted layers only a few segments long (*Israelachvili, 2011*). We estimated the length ($L$) of the brush as (*Rubinstein and Colby, 2003*):

$$L \sim N \Gamma^{\frac{1-\nu}{2\nu}} b^{\frac{1}{\nu}} \qquad (10)$$

Where $N$ is the number of monomers per grafted chain, $\nu$ is the Flory exponent, and $b$ is the Kuhn length of the grafted polymer. We used $b = 0.76$ nm based on literature measurements (*Waters et al., 2010*) and took $\nu \cong 0.588$, because aqueous salt solutions are good solvents for PEG (*Kawaguchi et al., 1997*). Lastly, we estimated the number of monomers per chain by assuming the number of monomers is approximately equation to the number of Kuhn segments and the relationship between the radius of gyration, the Kuhn length and the number of Kuhn segments (*Rubinstein and Colby, 2003*): $N \sim \left(\frac{R_g}{b}\right)^{\frac{1}{0.588}} \sim 13$. We therefore estimate that $L \sim 6.4$ nm.

The Alexander–de Gennes approximation assumes a step profile for the volume fraction of the grafted polymer ($\phi$). We can estimate this using the following equation (*Rubinstein and Colby, 2003*):

$$\phi \approx \begin{cases} \left(\Gamma b^2\right)^{\frac{3\nu-1}{2\nu}} & for\ z < L \\ 0 & for\ z > L \end{cases} \tag{11}$$

Where $z$ is the distance from the bare particle surface. Using the same approximations as above we find $\phi \approx 0.43$.

## Western blot of luminal contents

The 30-µm-filtered small intestinal luminal fluid was reduced in sample buffer with 100 mM dithio-threitol (DTT) at 95°C for 5 min (the luminal fluid was diluted 10-fold in the sample buffer). Gel electrophoresis was then run on 4–15% SDS/PAGE gels. The transfer was performed using wet electroblotting to a nitrocellulose membrane. For detection of MUC2, the primary antibody was diluted 1:1000 (MUC2 polyclonal antibody, rabbit host, Biomatik, Wilmington, DE, USA) as a 1:10,000 in Odyssey blocking buffer (Li-Cor, Lincoln, NE, USA) with 0.2% Tween 20. The secondary antibody (Li-Cor IRDye 800CW Goat Anti-Rabbit IgG, Li-Cor) was diluted 1:10,000. For the detection of IgG and IgM, 1:10,000 dilutions of Li-Cor IRDye 800 CW Goat Anti-Mouse IgG and Li-Cor IRDye 800CW Goat Anti-Mouse IgM were used respectively. For detection of IgA, a 1:10,000 dilution of SouthernBiotech Goat Anti-Mouse IgA-unlabeled was used as the primary and a 1:10,000 dilution of Li-Cor IRDye 800CW Donkey Anti-Goat IgG was used as the secondary. All membranes were visualized using a Li-Cor Odyssey scanner.

## Gel permeation chromatography

We used a Malvern OMNISEC RESOLVE connected to two Malvern A6000M columns (Malvern, Westborough, MA, USA) equilibrated with 1x PBS with 0.02% sodium azide, flow rate: 0.75 mL/min. For detection of the polymers, the OMNISEC REVEAL was used with a refractometer, UV detector, dual-angle light scattering detector, and a capillary viscometer. Luminal contents were 0.45 µm filtered as described above, then diluted 10-fold in the running buffer (1x PBS with 0.02% sodium azide) before injection into the system. Prior to injection, samples were kept on the autosampler at 4°C.

## Synthesis of PEG-coated particles

We amended a previously published protocol (*Maisel et al., 2015a*) to synthesize PEG-coated particles; briefly, 2 mL of 1 µm fluorescent carboxylic-acid-terminated polystyrene beads (Fluoro-Spheres, Invitrogen, Thermo Fisher Scientific) at 2% v/v with 2 mM NaN$_3$ were rinsed at 3900 g for 40 min using a centrifugal filter (Millipore Amicon Ultra-4 mL 100 K MWCO). Particles were removed from the filter using 4 mL of a solution of 15 mg/mL 1-Ethyl-3-(3-dimethylaminopropyl)carbodiimide (EDC, Sigma-Aldrich) and 15 mg/mL N-hydroxysuccinimide (NHS, Aldrich), an excess concentration of NH$_2$-PEG-OMe (5 kDa, Creative PEGworks, Chapel Hill, NC, USA) in 1 mL increments using 100 mM borate buffer, pH 8.4. By an excess concentration of NH$_2$-PEG-OMe we mean ten-fold the concentration of PEG required to enter the polymer brush regime (see 'Estimation of coverage and length of grafted PEG layer' section for details of calculation). This solution was tumbled on a rotary tumbler for 4 hr at room temperature in a 15 mL falcon tube. Particles were washed three times to remove starting materials with 4 mL Milli-Q water in a centrifugal filter and re-suspended in 2 mL of Milli-Q water.

## Synthesis of PEG-coated particles with 'backfill'

12 mL of 1 µm fluorescent carboxylic-acid-terminated polystyrene beads at 2% v/v with 2 mM NaN$_3$ (FluoroSpheres 1 µm; 505/515, Invitrogen) were centrifuged to a pellet at 12,000 g for 10 min. Beads were pelleted and rinsed three times with Milli-Q water. To the final pellet of particles, 12 mL of a solution of 6 mM EDC (10 mg/mL; Sigma-Aldrich) and 5 mM Sulfo-NHS (1.08 mg/mL, ThermoFisher), with 50x excess of the number of chains needed to enter the brush regime (see 'Estimation of coverage and length of grafted PEG layer' for details of calculation) of NH$_2$-PEG-OMe (mPEG-Amine 2 kDa; mPEG-Amine 5 kDa; Creative PEGWorks) in 10x PBS, pH 7.4 (100 mM), was added. This

solution was tumbled on a rotary tumbler for 4 hr at room temperature. Tubes were vented every 30 min to release gas produced by the reaction. Particles were then pelleted and rinsed three times with Milli-Q water. The 12 mL sample was divided into four 3 mL aliquots for the remaining conditions. For condition without backfill, beads were quenched with 50 mM Tris pH 7.4 overnight at room temperature with slow tilt rotation prepared from 10x Tris-buffered saline with Tween 20, pH 7.5 (Sigma-Aldrich). For particles with backfill, the 3 mL aliquot was re-suspended in 50x excess of the number of chains needed to enter the brush regime (see 'Estimation of coverage and length of grafted PEG layer' for details of calculation) of $NH_2$-PEG-OMe (mPEG-Amine 350; mPEG-Amine 1 kDa; mPEG-Amine 5 kDa, Creative PEGWorks) in 100 mM PBS, pH 7.4 containing 6 mM EDC and 5 mM Sulfo-NHS for 4 hr before quenching overnight with 50 mM TRIS buffered Saline with Tween 20, pH 7.5. All beads were washed three times with Milli-Q water before suspending in 3 mL sterile filtered PBS, pH 7.4 with 1% BSA for storage.

## NMR of PEG-coated particles with 'backfill'

We took 400 µl of 2% w/v samples and lyophilized (~8 mg), then dissolved in deuterated chloroform (Cambridge Isotope Laboratories, Tewksbury, MA, USA) with 0.01% tetramethylsilane (Aldrich) immediately before measurement. Data were collected on a Varian Innova 600 MHz spectrometer without spinning, using a 45° pulse width and 1 s relaxation delay between scans. The concentration of PEG in each sample was determined by integrating the singlet at 3.64 pm and normalizing the integral to TMS internal standard at 0.0 ppm.

## Zeta potential measurements on PEG-coated particles with 'backfill'

Each particle solution was 0.1 mg/mL of particles in 1 mM KCl. Measurements were done on a Brookhaven NanoBrook ZetaPALS Potential Analyzer (Brookhaven Instruments Corporation, Holtsville, NY, USA). Three trials were done where each trial was 10 runs and each run was 10 cycles. Values reported are the average zeta potential for the 30 runs.

## Estimate of Weissenberg number for small intestine

The Weissenberg number (Wi), which weighs the relative contributions of elastic and viscous forces, can be written as (*Arratia et al., 2005*):

$$Wi = \dot{\gamma}\lambda \tag{12}$$

Where $\dot{\gamma}$ is the shear rate (in $s^{-1}$) and $\lambda$ is the fluid relaxation time (in $s$). The shear rate in the human small intestine during peristaltic contractions has been estimated as $\dot{\gamma} \sim 29\ s^{-1}$ (*Takahashi, 2011*). For dilute aqueous polymeric solutions of polyacrylamide with MWs ranging from $10^4$ to $10^7$ Da, it has been found that $\lambda$ = 0.009 to 0.45 s, with the relaxation time increasing with MW as $\lambda \propto MW^{2/3}$ (*Arratia et al., 2009*). Using these values, we can estimate the Weissenberg number to be Wi ~ 0.3 to 10.

## Acknowledgements

This work was supported in part by DARPA Biological Robustness in Complex Settings (BRICS) contract HR0011-15-C-0093, Army Research Office (ARO) Multidisciplinary University Research Initiative (MURI) contract #W911NF-17-1-0402, the Jacobs Institute for Molecular Engineering for Medicine, and an NSF Graduate Research Fellowship DGE-144469 (to APS). We acknowledge Michael Porter, Joong Hwan Bahng, Jacob Barlow, Zhen-Gang Wang, Julia Kornfield, David Tirrell, Justin Bois, and Greg Donaldson for useful discussions; the Beckman Institute Biological Imaging Facility, the Broad Animal Facility, and the Church Animal Facility for experimental resources; Jennifer Costanza, Taren Thron, the Caltech Office of Laboratory Animal Resources, and the veterinary technicians at the California Institute of Technology for technical support; Joanne Lau for assistance with Western blot measurements; Emily Wyatt for assistance with zeta potential measurements; the Mazmanian laboratory for providing Rag1KO mice; the Eugene Chang Lab (University of Chicago) for providing the initial breeding pairs for the MUC2KO colony and Leonard H Augenlicht at the Department of Oncology of Albert Einstein Cancer Center for providing the original MUC2KO line to them; and Natasha Shelby for contributions to writing and editing this manuscript.

## Additional information

### Competing interests

Rustem F Ismagilov: The technology described in this publication is the subject of provisional patent application 62/696,743, filed by Caltech on 7/11/18. The other authors declare that no competing interests exist.

### Funding

| Funder | Grant reference number | Author |
| --- | --- | --- |
| Defense Advanced Research Projects Agency | HR0011-15-C-0093 | Rustem F Ismagilov |
| Army Research Office | W911NF-17-1-0402 | Rustem F Ismagilov |
| National Science Foundation | DGE-1144469 | Asher Preska Steinberg |
| Jacobs Institute for Molecular Engineering for Medicine | | Rustem F Ismagilov |

The funders had no role in study design, data collection and interpretation, or the decision to submit the work for publication.

### Author contributions

Asher Preska Steinberg, Conceptualization, Resources, Datacuration, Software, Formal analysis, Funding acquisition, Validation,Investigation, Visualization, Methodology, Writing—original draft, Writing—reviewand editing, Co-designed all experiments and co-analyzed all experimentalresults; developed theoretical tools and performed all calculations; co-developed imaging analysis pipeline in ImageJ; developed computational toolsfor bootstrapping procedure; co-developed microscopy assay (Fig 1C-D;Co-performed, designed, and analyzed data from gavage experiments in Fig 1;performed, designed, and analyzed data from all ex vivo SI aggregationexperiments in Figs 2, 3, 5-7; performed, designed, and analyzed data from allGPC measurements in Figs 3, 5-7, and Tables 1-7; performed, designed, andanalyzed data from all in vitro PEG aggregation experiments in Fig 4D, Fig 4 - supplements2-3, and with dietary fiber in Fig 7A; developed a computational approach fortheoretical calculations in 4H and 4I and performed all calculations;performed, designed, and analyzed data from Western blots in Figs 5E, 6E, Fig 6- supplements 1-2; helped supervise animal husbandry of MUC2KO colony;performed animal husbandry for WT mice on autoclaved diets in Fig 6; performedanimal husbandry for mice on pectin and Fibersol-2 diets in Fig 7; performed,designed, and analyzed all zeta potential measurements in Table 8; performed pHmeasurements on luminal fluid in Fig 4 - supplement 1; co-interpreted results.; Sujit S Datta, Conceptualization, Investigation,Methodology, Writing—review and editing, Conceived and co-planned the project;initially observed the aggregation phenomenon; co-designed and co-analyzedpreliminary experiments; performed preliminary ex vivo and in vitro aggregationexperiments; co-developed microscopy assay used in Fig 1C and 1D; developed exvivo/in vitro aggregation assay used in Figs 2-7; co-developed approach to extractliquid fraction of murine intestinal contents; co-developed NMR protocol;organized transfer and initial set up of MUC2KO colony; co-interpreted results.; Thomas Naragon, Data curation, Software, Formal analysis, Methodology, Writing—original draft, Co-developed imaging analysis pipeline in ImageJ; co-analyzed ex vivo aggregation data in Fig 2; co-designed and co-analyzed preliminary ex vivo aggregation experiments with MUC2KO mice; provided useful advice on bootstrapping procedure; co-interpreted results.; Justin C Rolando, Data curation, Formal analysis, Investigation, Methodology, Writing—original draft, Developed protocol for NMR measurements on PEG-coated particles, Performed synthesis of particles, Performed NMR measurements in Table 8; Said R Bogatyrev, Investigation, Methodology, Writing—review and editing, Co-performed preliminary experiments; developed fluorescent laser scanning approach appearing in Fig 1A and 1B; Administered particles to mice inFig 1; co-developed approach to extract liquid fraction of murine intestinal contents; co-organized transfer and initial set up of MUC2KO colony; setup genotyping of MUC2KO mice; helped supervise animal husbandry of MUC2KO colony; helped with interpretation

of results.; Rustem F Ismagilov, Resources, Formal analysis, Supervision, Funding acquisition, Investigation, Methodology, Writing—original draft, Project administration, Writing—review and editing

### Author ORCIDs
Asher Preska Steinberg (iD) http://orcid.org/0000-0002-8694-7224
Sujit S Datta (iD) https://orcid.org/0000-0003-2400-1561
Thomas Naragon (iD) https://orcid.org/0000-0002-5373-4257
Justin C Rolando (iD) https://orcid.org/0000-0001-8948-319X
Said R Bogatyrev (iD) http://orcid.org/0000-0003-0486-9451
Rustem F Ismagilov (iD) http://orcid.org/0000-0002-3680-4399

### Ethics
Animal experimentation: All animal experiments were approved by the California Institute of Technology (Caltech) Institutional Animal Care and Use Committee (IACUC) under IACUC protocol #1691, and the U.S. Army's Animal Care and Use Review Office (ACURO) under ACURO protocols #DARPA-533.02 and #70905-LS-MUR.03. Mice were euthanized via $CO_2$ inhalation as approved by the Caltech IACUC in accordance with the American Veterinary Medical Association Guidelines on Euthanasia.

### Decision letter and Author response
Decision letter https://doi.org/10.7554/eLife.40387.029
Author response https://doi.org/10.7554/eLife.40387.030

## Additional files

### Supplementary files
• Transparent reporting form
DOI: https://doi.org/10.7554/eLife.40387.025

### Data availability
All data generated or analyzed during this study have been uploaded to Dryad (http://dx.doi.org/10.5061/dryad.kd1qt0p).

The following dataset was generated:

| Author(s) | Year | Dataset title | Dataset URL | Database and Identifier |
|---|---|---|---|---|
| Asher Preska Steinberg | 2018 | Data from: High-molecular-weight polymers from dietary fiber drive aggregation of particulates in the murine small intestine | http://dx.doi.org/10.5061/dryad.kd1qt0p | Dryad Digital Repository, 10.5061/dryad.kd1qt0p |

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
