## [Decision Letter]

[**Editorial note:** This article has been through an editorial process in which the authors decide how to respond to the issues raised during peer review. The Reviewing Editor's assessment is that all the issues have been addressed.]

Thank you for submitting your article "High-molecular-weight polymers from dietary fiber drive aggregation of particulates in the murine small intestine" for consideration by *eLife*. Your article has been reviewed by two peer reviewers, and the evaluation has been overseen by a Reviewing Editor and Arup Chakraborty as the Senior Editor. The following individual involved in review of your submission has agreed to reveal his identity: Paulo Arratia (Reviewer #2). Reviewer #1 remains anonymous.

The Reviewing Editor has highlighted the concerns that require revision and/or responses, and we have included the separate reviews below for your consideration. If you have any questions, please do not hesitate to contact us.

Summary:

This paper addresses, through experiment and theory, the role of high molecular

weight polymers, such as those from dietary fibre, in inducing aggregation of small particles. This aggregation is hypothesized to arise from depletion interactions of the kind well-known in colloidal science. A variety of experimental techniques, including an ex-vivo method, was used to test this hypothesis, and the results were interpreted in the context of a synthesis of results from polymer and colloidal physics concerning depletion interactions, polymer brushes, and related phenomena.

Major concerns:

The reviewers raised several significant concerns. First, the issue of the change in intraluminal pH along the length of the SI should be addressed and further control experiments considered. Second, the connection between the results on model polymer solutions and the considerably more complex SI environment could be improved. Third, the authors have not accounted for the effects of fluid flow and indeed any of the complex rheology expected in the SI.

Separate reviews (please respond to each point):

*Reviewer #1:*

This paper reports on the spontaneous aggregation of particulate matter in the small intestine (SI), highlighting the role played by high-molecular-weight polymers from dietary fiber in driving the aggregation. A provocative corollary to this finding is that it may be possible to control ("tune") the aggregation using suitably selecting the molecular weights of dietary polymers. The proposed mechanism is in depletion interactions that are well known in colloidal chemistry and polymer physics. Given the common perception that aggregation of particulates such as food granules or small bacteria or pathogens is important to human health, the paper is of potential interest to a wider class of researchers beyond the food industry, and is appropriate for a journal such as *eLife*. The paper has an extensive set of results from both in vivo studies and ex vivo models with good statistics. An impressively wide range of physical and biochemical methods are used (NMR for verifying PEGylation; fluorescence imaging; gel permeation chromatography to analyze fluids; zeta potential measurements, etc) as well as a comprehensive description of nanoparticle synthesis, and animal studies. The animal studies are interesting and comprehensive, with a systematic approach that is generally well-conceived. The central finding, namely that high-molecular-weight polymers promote aggregation in the SI, is certainly reasonable. The molecular level understanding of this process claimed by the authors, while reasonable, has not been established in the reported studies.

Summary of substantive concerns:

1) A surprising omission in the paper is not to comment on the change in intraluminal pH throughout the length of the small intestine. Entering from the extremely low pH in the stomach, particulates encounter pH ranging from about pH 6 in the duodenum, increasing to about pH 7.3 or 7.4 in the terminal ileum. Given that pH is known to strongly modulate polymer aggregation the lack of a complete discussion in the paper is a serious omission. Control experiments in which the aggregation is studied in vitro over a pH range from about 6 to pH 7.4 could show a similar aggregation, compared to studies conducted at a single average pH set by HBSS. This does not detract from one conclusion of this paper – namely the role played by high-molecular-weight polymers in aggregation. But they need to discuss and account for the change in pH in their analysis.

2) The paper stitches together results from a broad range of equilibrium theoretical models which generally apply in different physical regimes, and are not quite compatible. The concern is that the extensive studies span parameter ranges in which theoretical models do not comfortably overlap. The Asakura-Oosawa potential is applicable in relatively dilute regimes, as is the equation for the radius of gyration with an exponent w.r.t molecular weight of 0.5. The depletion potential from de Gennes (with Joanny and Liebler) which in the semi-dilute regime known to destabilize colloidal particle solutions. These models are required to obtain the polymer overlap concentration cp* to estimate the osmotic pressure. The use of Flory-Huggins theory for mixing, if extended to aggregating systems, will require higher order terms in the volume fraction to be consistent. There is an attempt to address the need for higher order terms by using short range repulsion attributed to deformation of polymer graft layers. The use of each of these models is defensible, and the authors have done a good job of systematically laying out the assumptions involved. But partly as a result of this melding of different theoretical models the connection between the relatively clean measurements of model polymer solutions, and the enormously more complicated SI studies is rather shaky. The paper appears to be stretching points to make the connections. The analysis of at least the model polymer systems is likely to be robust. There is less confidence in that the analysis could carry over to the SI, where (as stated above) the parameters are pH dependent, strongly modulating the depletion interactions, and perhaps influencing the deformation of the polymer grafts.

To reiterate, the experiments, both in vivo and ex vivo, are both interesting and comprehensive, systematic and well-conceived. The analysis of the ex vivo experiments is a bit of a kludge, but likely to hold. The extension of such an analysis to the SI is not well justified. Given all this complexity, a lack of discussion of the role of pH in modulating colloidal polymer behavior is rather striking.

3) In the discussion of aggregation kinetics, particular in animal studies in the SI, it is not clear if the time scales are long enough to justify the use of essentially quasi-equilibrium statistical mechanical models. Perhaps the change in pH experienced by particulate matter during transport along the SI is sufficiently slow, but some discussion of the time scales would be helpful. (Experiments on model systems presumably took place over long enough time-scales that this may not be an issue.)

Minor Comments

1) Equations should be numbered.

2) The lack of a role of mucin (MUC2) may seem at first sight to be surprising, but MUC2 is designed for a pH triggered transition and is likely to be in different phases along different locations of the small intestine. Above about pH4 or pH5, mucin is likely to be in sol phase anyway; so perhaps the studies with the MUC2 knockout mice are informative but perhaps to be expected.

3) Figure 7, the chromatograms are plotted with the refractive index (RI) in mV. While the output of the RI detector may be in mV, the refractive index is a dimensionless scalar, and the axis should be labeled as such.

4) Perhaps this is a very minor issue, but how much is the analysis affected by throwing out the undiluted samples from pectin-fed mice?

*Reviewer #2:*

This is an interesting article that applies concepts from polymer physics and colloidal sciences to understand a relevant biological, namely particle aggregation in the smaller intestine (SI). The main result appears to be that even non-interacting particles can aggregate in the luminal fluid due to the presence of polymers of a rage of molecular weights and concentrations in the fluid. Importantly, the authors demonstrate that mucin (MUC2) and immunoglobulins are not strictly necessary for the formation of aggregates, although the experiments are preliminary on that front (the combination of them was not studied). Furthermore, the authors show that a diet rich in relatively long polymer chains (Pectin) can lead to particle aggregation in mice compared to a diet of low MW polymers (Fibersol).

The data presented by the authors are in my opinion convincing enough, but I do have a few comments. I am concerned that the authors do not address the effect of fluid flow. Is it because it is too slow? The issue here is that it has long been known that the combined effects of flow (i.e. velocity gradients) and fluid elasticity (i.e. polymers) can lead to particle aggregation even for non-interacting particles. In other words, the aggregation observed in vivo can also be the result of elastic stresses from the polymeric fluids aiding the aggregation process. It would make sense that longer chains (usually associated with higher levels of elasticity) would lead to larger aggregates. Shear-thinning viscosity also plays a role and the phenomena is thus nonlinear and non-monotonic. It would be good for the authors to at least comment on this; I do understand that their in vitro experiments do not include flow.

The data in Figure 4 is convincing, but I would be good if the authors could normalize the polymer concentration by the overlap concentration (c*). That way one could gain a better understanding on the effects of polymer MW and type by focusing on a particular normalized concentration (c/c*). Would this normalization be sufficient to observe collapse in the MW curves?

Additional data files and statistical comments

I would be very interested to know the rheology of the polymeric solutions used in their studies. I must stress however that this is not a must.

---

## [Author Response]

Major concerns:The reviewers raised several significant concerns. First, the issue of the change in intraluminal pH along the length of the SI should be addressed and further control experiments considered. Second, the connection between the results on model polymer solutions and the considerably more complex SI environment could be improved. Third, the authors have not accounted for the effects of fluid flow and indeed any of the complex rheology expected in the SI.

We thank the editors for highlighting the key points that the reviewers have raised. We have done further control experiments to address the issue of pH. We have modified the text of the manuscript to address the connection between the model polymer solutions and the SI environment. Lastly, we have modified the text of the manuscript to account for the effects of fluid flow and the complex rheology of the SI.

Separate reviews (please respond to each point):

Reviewer #1:

This paper reports on the spontaneous aggregation of particulate matter in the small intestine (SI), highlighting the role played by high-molecular-weight polymers from dietary fiber in driving the aggregation. A provocative corollary to this finding is that it may be possible to control ("tune") the aggregation using suitably selecting the molecular weights of dietary polymers. The proposed mechanism is in depletion interactions that are well known in colloidal chemistry and polymer physics. Given the common perception that aggregation of particulates such as food granules or small bacteria or pathogens is important to human health, the paper is of potential interest to a wider class of researchers beyond the food industry, and is appropriate for a journal such as eLife. The paper has an extensive set of results from both in vivo studies and ex vivo models with good statistics. An impressively wide range of physical and biochemical methods are used (NMR for verifying PEGylation; fluorescence imaging; gel permeation chromatography to analyze fluids; zeta potential measurements, etc) as well as a comprehensive description of nanoparticle synthesis, and animal studies. The animal studies are interesting and comprehensive, with a systematic approach that is generally well-conceived. The central finding, namely that high-molecular-weight polymers promote aggregation in the SI, is certainly reasonable. The molecular level understanding of this process claimed by the authors, while reasonable, has not been established in the reported studies.

We thank the reviewer for their feedback on the manuscript, and are encouraged by their kind comments on the broad interest of this work to the community, the scientific rigor of the work, and the appropriateness of the manuscript for *eLife*. We address the reviewer’s comments below.

Summary of substantive concerns:1) A surprising omission in the paper is not to comment on the change in intraluminal pH throughout the length of the small intestine. Entering from the extremely low pH in the stomach, particulates encounter pH ranging from about pH 6 in the duodenum, increasing to about pH 7.3 or 7.4 in the terminal ileum. Given that pH is known to strongly modulate polymer aggregation the lack of a complete discussion in the paper is a serious omission. Control experiments in which the aggregation is studied in vitro over a pH range from about 6 to pH 7.4 could show a similar aggregation, compared to studies conducted at a single average pH set by HBSS. This does not detract from one conclusion of this paper – namely the role played by high-molecular-weight polymers in aggregation. But they need to discuss and account for the change in pH in their analysis.

The reviewer brings up an excellent point, which has enabled us to clarify the role of pH in our manuscript. We have taken several actions to address this. Although there is existing literature in which the pH is measured along the gastrointestinal tract of both rats and mice, our animals were of different strains (and in some cases different species) and were kept on different diets, so we decided to conduct additional experiments in which we measured the pH along the gastrointestinal tract of the mice used in our studies. These measurements are included as a new figure, Figure 4—figure supplement 1.

We found that the pH of the buffer we had used in previous experiments (Hank’s balanced salt solution with a measured pH=7.6 ± 0.1) matched that which we measured in the lower small intestine (pH=7.5 ± 0.3) but was higher than the upper small intestine (pH=6.0 ± 0.1). Therefore, we conducted control experiments in which we measured aggregation in vitro in a buffer of pH = 6 with a matching ionic strength. These data have been included as Figure 4—figure supplement 2.

We have also made the following additions to the text:

Main text, subsection “Aggregation of PEG-coated particles in model polymer solutions shows complex dependence on the concentration and MW of polymers”:

“For PEG 1 MDa and 100 kDa solutions we found aggregates of similar sizes to those observed in the SI luminal fluid (Figure 4A-D). We did not detect any aggregation for the PEG 3350 Da solutions (Figure 4D). Because the pH is known to vary across different sections of the gastrointestinal tract and this could affect the observed aggregation behavior, we measured the pH in luminal fluid from the upper and lower small intestine (see Figure 4—figure supplement 1 and Materials and methods). We found that the upper small intestine (USI) luminal fluid was pH=6.0 ± 0.1and for the lower small intestine (LSI)pH=7.5 ± 0.3. For the HBSS used, pH=7.6 ± 0.1(See Materials and methods), which matches that of the LSI but not the USI. We therefore conducted the same in vitro experiment for PEG 1 MDa in phosphate buffered saline with pH=6.0 ± 0.1(Materials and methods and Figure 4—figure supplement 2). We found some differences in the aggregation, but the overall trends were similar to before.”

Materials and methods:

“In the 1 MDa PEG experiments conducted in phosphate buffered saline (PBS) with pH = 6 (Figure 4—figure supplement 2) the PBS solution was initially prepared with 138 mM sodium chloride, 7.5 mM monosodium phosphate dihydrate, 1.1 mM disodium phosphate heptahydrate, and deionized (DI) water (Milli-Q). The sodium chloride was added to ensure that the ionic strength matched that of Hank’s balanced salt solution. The pH was then measured using an Orion 2-Star Benchtop pH Meter (Thermo Scientific) with an Orion 9110DJWP Double Junction pH electrode (Thermo Fisher Scientific) after first calibrating the instrument using the following reference standard buffers: pH = 10 (VWR BDH5078-500 mL), pH = 7 (VWR BDH5046-500 mL), and pH = 4 (VWR BDH5024-500 mL). The pH of the solution was then adjusted to pH = 6 using 1 M NaOH in DI water.”

Materials and methods:

“pH measurements of luminal fluid. Pooled samples of luminal fluid were collected from each section (stomach, upper small intestine, lower small intestine, cecum, and colon) and 30 µm-filtered as described in “Collection of Luminal fluid” (with use of the same protease inhibitor cocktail). Samples were collected from two separate groups of 2-month-old B6 male mice on a standard chow diet. Each group had three mice. Because there was only ~25 µL of luminal fluid from the colons of each group we did not 30 µm-filter the colonic fluid as there was concern all the fluid would be retained by the filter. The colonic contents were simply spun down at 17 kG at 4 °C for 1 h to separate the liquid and solid portions of the contents. Then the supernatant (luminal fluid) was collected. Measurements were done using an Orion 2-Star Benchtop pH Meter. The instrument was first calibrated with three reference standard buffers: pH = 10 (VWR BDH5078-500 mL), pH = 7 (VWR BDH5046-500 mL), and pH = 4 (VWR BDH5024-500 mL). Measurements were conducted at T = 25 °C. There was at least 100 µL of sample from each section except for the stomach sample from one group of mice and from colon samples from both groups. Measurements were conducted with both a standard pH electrode (Orion 9110DJWP Double Junction pH Electrode) and a micro pH electrode (Orion 9810BN Micro pH Electrode, Thermo Fisher Scientific). This was done because the standard electrode is only accurate for samples with volumes of 200 µL whereas the micro electrode is accurate for samples as small as 0.5 µL in volume. The results are consistent with other results for rodents (101,102) with the exception of a study conducted with mice of a different gender, strain, and fed an 18% protein diet (103).

For the pH measurement of HBSS, the pH was measured with both the standard and micro pH electrodes, and three technical replicates were done with each probe. The value for the pH reported in the main text is the average of all six measurements.”

2) The paper stitches together results from a broad range of equilibrium theoretical models which generally apply in different physical regimes, and are not quite compatible. The concern is that the extensive studies span parameter ranges in which theoretical models do not comfortably overlap. The Asakura-Oosawa potential is applicable in relatively dilute regimes, as is the equation for the radius of gyration with an exponent w.r.t molecular weight of 0.5. The depletion potential from de Gennes (with Joanny and Liebler) which in the semi-dilute regime known to destabilize colloidal particle solutions. These models are required to obtain the polymer overlap concentration cp* to estimate the osmotic pressure. The use of Flory-Huggins theory for mixing, if extended to aggregating systems, will require higher order terms in the volume fraction to be consistent. There is an attempt to address the need for higher order terms by using short range repulsion attributed to deformation of polymer graft layers. The use of each of these models is defensible, and the authors have done a good job of systematically laying out the assumptions involved. But partly as a result of this melding of different theoretical models the connection between the relatively clean measurements of model polymer solutions, and the enormously more complicated SI studies is rather shaky. The paper appears to be stretching points to make the connections. The analysis of at least the model polymer systems is likely to be robust. There is less confidence in that the analysis could carry over to the SI, where (as stated above) the parameters are pH dependent, strongly modulating the depletion interactions, and perhaps influencing the deformation of the polymer grafts.

We thank the reviewer for bringing up these insightful points, as it has helped us clarify the theory and claims presented in the manuscript. We understand the reviewer’s comment that the studies span both the dilute and semi-dilute regimes, where different theories are applicable. This choice was made because based on our gel permeation chromatography measurements, we found polymers at concentrations in both the dilute and semi-dilute regimes. We would like to note that similar crossover equations have been used to describe entropic interactions in model polymer/colloid solutions that span the dilute and semi-dilute regimes and we have provided a reference. In addition, we do estimate the polymer overlap concentrations using literature values.

We also completely agree with the reviewer’s point that this analysis might not carry over to the small intestine. We would like to note that we have not attempted to apply this theory to the ex vivodata but only to the model polymers. We simply want to draw attention to the similarities in the behaviors observed in the model system to those observed in the ex vivoexperiments with small intestinal luminal fluid. We have made the following additions to the text to address this:

Main text, subsection “Aggregation of PEG-coated particles in model polymer solutions shows complex dependence on the concentration and MW of polymers”:

“Similar crossover equations have been found to adequately describe experimentally observed depletion aggregation in polymer-colloid mixtures where the polymer concentration spans the dilute and semi-dilute regimes.”

Main text, subsection “Polymers in the diet control aggregation of PEG-coated particles in a manner consistent with depletion type interactions”:

“Due to the high polydispersity and complex chemical composition of SI luminal fluid as measured by GPC, it is unfeasible to apply the same theoretical analysis as was done in Figure 4 to these data. We can, however, note that visually the behavior is qualitatively consistent with the depletion-type interactions found in simple PEG solutions in Figure 4.”

Discussion section, paragraph two:

“More work needs to be done to understand the underlying mechanism, but surprisingly the observed aggregation behavior in the SI luminal fluid from mice fed dietary polymers of different sizes is qualitatively consistent with the aggregation behavior in simple PEG solutions, where aggregation is driven by depletion interactions. Overall, this suggests a simple dietary method for controlling aggregation in the gut. It will be important to extend this work to microbes and other particles commonly found in the gut and to measure the relative contributions of polymer-driven aggregation and chemical-driven aggregation. We note that mucins and immunoglobulins are polymers that can also self-associate into structures of very high MW (78,87,88), suggesting that they could cause aggregation via both physical and chemical mechanisms. Interestingly, during the review of this manuscript, a study was published with in vitrowork done using model buffer solutions of mucins, DNA, and other biopolymers further implying that aggregation of bacteria by host-polymers can be depletion-mediated (89).”

*To reiterate, the experiments, both* in vivo *and* ex vivo*, are both interesting and comprehensive, systematic and well-conceived. The analysis of the* ex vivo *experiments is a bit of a kludge, but likely to hold. The extension of such an analysis to the SI is not well justified. Given all this complexity, a lack of discussion of the role of pH in modulating colloidal polymer behavior is rather striking.*

We thank the reviewer for these comments and for acknowledging the novelty of the work as well as our systematic approach. We have addressed the issues of pH and the analysis of the ex vivosmall intestine data as described under points 1 and 2 above.

3) In the discussion of aggregation kinetics, particular in animal studies in the SI, it is not clear if the time scales are long enough to justify the use of essentially quasi-equilibrium statistical mechanical models. Perhaps the change in pH experienced by particulate matter during transport along the SI is sufficiently slow, but some discussion of the time scales would be helpful. (Experiments on model systems presumably took place over long enough time-scales that this may not be an issue.)

We thank the reviewer for bringing this to our attention, as it has allowed us to clarify this point in the manuscript. We recognize that the statistical mechanical models we applied are equilibrium models, and thus should not yield exact predictions for our experiments, which were not taken to equilibrium. Our intention was to describe both approximate predictions for what would happen at equilibrium and describe the effect polymer viscosity would have on hindering aggregation on the short timescales of our experiments. We also have not untangled the interplay between kinetics and thermodynamics in our experimental setup, and our analysis is limited by that. The reviewer also brings up an important point about which timescale is relevant for understanding what happens in the small intestine. We chose the short experimental timescale used because we wanted to understand the initial formation of aggregates before aggregation is influenced by mechanical forces such as shear due to peristaltic mixing and the transit of food through the SI. The transit time for food through the SI can be as short as ~80 minutes in humans and ~60 minutes in mice. During fasting, the migrating motor complex cycle first consists of a period of quiescence for ~30-70 minutes followed by contractions that induce peristaltic mixing. We therefore expect that the aggregation behavior will be more complex on these longer timescales. We have made the following additions to the text to address these points:

Results subsection “PEG-coated particles aggregate in fluid from the murine small intestine”:

“We found that, despite the PEG coating and low-volume fraction, aggregates of particles formed in 5-10 min (Figure 2*A-D*), a timescale much shorter than the transit time for food through the SI, which can be as short as ~80 min in healthy humans (39) and ~60 min in mice (44). On longer timescales, peristaltic mixing could also play a role (39); during fasting, the migrating motor complex (MMC) cycle first consists of a period of quiescence for ~30-70 min, followed by a period of random contractions, then by 5 to 10 minutes in which contractions occur at 11-12 counts per minute (cpm) in the duodenum and 7-8 cpm in the ileum. After eating, MMC is substituted with intermittent contractions in the SI and waves can occur at a frequency of 19-24 cpm in the distal ileum 1-4 h later. We therefore chose to focus on aggregation at short timescales (~10 min) because we sought to understand the initial formation of aggregates before aggregation is influenced by mechanical forces such as shear due to peristaltic mixing and the transit of food.”

Results, subsection “Aggregation of PEG-coated particles in model polymer solutions shows complex dependence on the concentration and MW of polymers”

“We again chose to look at aggregation on short timescales (after ~10 min) because we sought to understand the initial formation of aggregates; in the SI, on longer timescales, aggregation will likely also be influenced by mechanical forces such as shear due to peristaltic mixing and the transit of food.”

and:

“Overall, though our system is not at equilibrium at these short timescales, we found trends consistent with what has been observed in the literature for depletion interactions with sterically stabilized particles (49–57).”

and:

“Because our system has not reached equilibrium, in this case the non-monotonic dependence of aggregation on polymer concentration for long polymers is due to a complex interplay between thermodynamics and kinetics (which we have not untangled). However, both the dependence of diffusivity (Figure 4I) and the equilibrium prediction of inter-particle minima (Figure 4H)on polymer concentration suggest that we should expect a decrease in aggregation at high polymer concentrations.”

Minor Comments1) Equations should be numbered.

We thank the reviewer for pointing this out as this was an oversight. The equation labels were not obvious and we have re-formatted to make them more clearly visible.

2) The lack of a role of mucin (MUC2) may seem at first sight to be surprising, but MUC2 is designed for a pH triggered transition and is likely to be in different phases along different locations of the small instestine. Above about pH4 or pH5, mucin is likely to be in sol phase anyway; so perhaps the studies with the MUC2 knockout mice are informative but perhaps to be expected.

We thank the reviewer for pointing this out. It is known that MUC2 can form aggregates at pH 6.2 with the presence of high concentrations of calcium. The pH we have measured in the upper small intestine is pH=6.0 ± 0.1and in the lower small intestine is pH=7.5 ± 0.3. Therefore it may be possible that MUC2 forms aggregates or precipitates out of solution in the upper small intestine, but this is unlikely in the lower small intestine. We have included the following additions to the text to address this:

Subsection “MUC2 may play a role in the aggregation of PEG-coated particles, but is not required for aggregation to occur”:

“It is known that in the presence of Ca^2+^ and at pH≤6.2, MUC2 can form aggregates or precipitate out, but it is soluble without Ca^2+^ or at higher pH (78). Our measurements of the pH throughout the SI suggest that it is possible that MUC2 precipitates out in the upper small intestine; however, because it is unclear how much Ca^2+^ is in the lumen of the upper small intestine, there could be soluble MUC2 in the upper small intestine. Additionally, the literature suggests that, based on the pH, there should be soluble MUC2 in the lower small intestine. We therefore tested if MUC2 drives aggregation in both the upper and lower small intestine.”

3) Figure 7, the chromatograms are plotted with the refractive index (RI) in mV. While the output of the RI detector may be in mV, the refractive index is a dimensionless scalar, and the axis should be labeled as such.

We thank the reviewer for pointing this out. We have changed the labels in Figures 3, 5, 6, and 7 to read “Differential Refractive Index” for differential refractive index and removed the units.

4) Perhaps this is a very minor issue, but how much is the analysis affected by throwing out the undiluted samples from pectin-fed mice?

We thank the reviewer for bringing this up. We would like to note that only one of the undiluted samples (from the 30-µm-filtered lower small intestine) from the pectin-fed mice was not used. In all other sets of samples (30- and 0.45-µm-filtered upper small intestine and 0.45-µm-filtered lower small intestine) the undiluted samples were used. Given that the undiluted 30-µm-filtered lower small intestine sample appeared to have a similar consistency to a viscous gel, we did not use it in our quantitative experiments because we could not guarantee that the particles would have mixed with the sample appropriately to conduct the experiment.

Reviewer #2:

This is an interesting article that applies concepts from polymer physics and colloidal sciences to understand a relevant biological, namely particle aggregation in the smaller intestine (SI). The main result appears to be that even non-interacting particles can aggregate in the luminal fluid due to the presence of polymers of a rage of molecular weights and concentrations in the fluid. Importantly, the authors demonstrate that mucin (MUC2) and immunoglobulins are not strictly necessary for the formation of aggregates, although the experiments are preliminary on that front (the combination of them was not studied). Furthermore, the authors show that a diet rich in relatively long polymer chains (Pectin) can lead to particle aggregation in mice compared to a diet of low MW polymers (Fibersol).

We thank the reviewer for their feedback on the manuscript, and are encouraged by their positive comments on the broad interest of this work to the community. We address the reviewer’s comments in detail below.

The data presented by the authors are in my opinion convincing enough, but I do have a few comments. I am concerned that the authors do not address the effect of fluid flow. Is it because it is too slow? The issue here is that it has long been known that the combined effects of flow (i.e. velocity gradients) and fluid elasticity (i.e. polymers) can lead to particle aggregation even for non-interacting particles. In other words, the aggregation observed in vivo can also be the result of elastic stresses from the polymeric fluids aiding the aggregation process. It would make sense that longer chains (usually associated with higher levels of elasticity) would lead to larger aggregates. Shear-thinning viscosity also plays a role and the phenomena is thus nonlinear and non-monotonic. It would be good for the authors to at least comment on this; I do understand that their in vitro experiments do not include flow.

We thank the reviewer for bringing up these excellent points, as it has allowed us to clarify them in our manuscript. The reviewer is certainly correct that the combined effects of flow and fluid elasticity as well as shear-thinning viscosity could contribute to aggregation in vivo. We have provided some estimates of what the contributions of some of these factors may be in vivo. We have made the following additions to the text to address this:

Discussion paragraph two:

“In vivo, it will also be important to consider the effects of flow, as it has been shown that flow in non-Newtonian fluids can induce particle aggregation (90–92). In particular, studies have suggested that the combination of flow and polymer elasticity can lead to aggregation (93) and that shear thinning viscosity can influence aggregation as well (94). In our work, we neglected flow effects for simplicity and thus our findings are most applicable to the initial formation of aggregates before aggregation is influenced by mechanical forces due to peristaltic mixing and the transit of food. A rudimentary estimate of the Weissenberg number (see *Materials and methods)*, whichweighs the contributions of elastic and viscous forces, yields Wi ~ 0.3 to 10, suggesting that elasticity-induced effects may play a role in the SI and will be an important direction to pursue in follow-up studies. If flow-induced clustering does occur in vivo, the literature suggests it would aid in the process, perhaps enhancing particle aggregation.”

Materials and methods:

**“**Estimate of Weissenberg number for small intestine.The Weissenberg number (Wi), which weighs the relative contributions of elastic and viscous forces, can be written as (108):

Wi=γ˙λ#Eq.12

Where γ˙ is the shear rate (in s-1) and *λ* is the fluid relaxation time (in *s*). The shear rate in the human small intestine during peristaltic contractions has been estimated as γ˙~29s-1 (109). For dilute aqueous polymeric solutions of polyacrylamide with MWs ranging from 10^4^ to 10^7^ Da, it has been found that λ=0.009 to 0.45 s, with the relaxation time increasing with MW as λαMW2/3(110). Using these values, we can estimate the Weissenberg number to be Wi ~ 0.3 to 10.”

The data in Figure 4 is convincing, but I would be good if the authors could normalize the polymer concentration by the overlap concentration (c*). That way one could gain a better understanding on the effects of polymer MW and type by focusing on a particular normalized concentration (c/c*). Would this normalization be sufficient to observe collapse in the MW curves?

The reviewer makes an excellent point. The normalization is indeed sufficient to collapse the experimental data presented in Figure 4D. We have included a re-normalized version of Figure 4D as a supplemental figure to Figure 4 (Figure 4—figure supplement 3).

Subsection “Aggregation of PEG-coated particles in model polymer solutions shows complex dependence on the concentration and MW of polymers”:

“Interestingly, we found that the curves for the long polymers in Figure 4D could be collapsed by normalizing the polymer concentration by the overlap concentration (which denotes the transition between the dilute to semi-dilute polymer concentration regimes) for each respective polymer solution (Figure 4—figure supplement 3).”

Additional data files and statistical commentsI would be very interested to know the rheology of the polymeric solutions used in their studies. I must stress however that this is not a must.

The reviewer makes a great point. These are experiments we plan to undertake in follow-up studies.

We would like to thank all of the reviewers for taking the time to carefully read the manuscript. Their insightful suggestions have helped us to further strengthen the manuscript, and we hope that the revised manuscript is now suitable for publication.